# Two *de novo* GluN2B mutations affect multiple NMDAR-functions and instigate severe pediatric encephalopathy

**Shai Kellner[1], Abeer Abbasi[1], Ido Carmi[1], Ronit Heinrich[1], Tali Garin-Shkolnik[2], Tova Hershkovitz[3], Moshe Giladi[4], Yoni Haitin[4], Katrine M Johannesen[5,6], Rikke Steensbjerre Møller[5,6], Shai Berlin[1]***

[1]Department of Neuroscience, Ruth and Bruce Rappaport Faculty of Medicine, Technion-Israel Institute of Technology, Haifa, Israel; [2]Clalit health services, Tel Aviv, Israel; [3]Genetics Institute, Rambam medical center, Haifa, Israel; [4]Department of Physiology and Pharmacology, Sackler Faculty of Medicine, Tel Aviv University, Tel Aviv, Israel; [5]Department of Epilepsy Genetics and Personalized Treatment, the Danish Epilepsy Centre, Dianalund, Denmark; [6]Institute for Regional Health Services, University of Southern Denmark, Odense, Denmark

**Abstract** The N-methyl-D-aspartate receptors (NMDARs; GluNRS) are glutamate receptors, commonly located at excitatory synapses. Mutations affecting receptor function often lead to devastating neurodevelopmental disorders. We have identified two toddlers with different heterozygous missense mutations of the same, and highly conserved, glycine residue located in the ligand-binding-domain of *GRIN2B*: G689C and G689S. Structure simulations suggest severely impaired glutamate binding, which we confirm by functional analysis. Both variants show three orders of magnitude reductions in glutamate $EC_{50}$, with G689S exhibiting the largest reductions observed for *GRIN2B* (~2000-fold). Moreover, variants multimerize with, and upregulate, GluN2B*wt*-subunits, thus engendering a strong dominant-negative effect on mixed channels. In neurons, overexpression of the variants instigates suppression of synaptic GluNRs. Lastly, while exploring spermine potentiation as a potential treatment, we discovered that the variants fail to respond due to G689's novel role in proton-sensing. Together, we describe two unique variants with extreme effects on channel function. We employ protein-stability measures to explain why current (and future) LBD mutations in GluN2B primarily instigate Loss-of-Function.

**\*For correspondence:** shai.berlin@technion.ac.il

**Competing interests:** The authors declare that no competing interests exist.

## Introduction

N-methyl-D-aspartate receptors (NMDARs or GluNRs) are excitatory glutamate receptors found throughout the brain, playing critical roles in neuronal development, synaptogenesis, plasticity, and in most processes of learning and memory (*Paoletti et al., 2013*; *Lau and Zukin, 2007*). The hetero-tetrameric receptor is assembled from seven different *GRIN* genes (*Glutamate Receptor, Ionotropic, NMDA-kind*) (*Paoletti et al., 2013*; *Hansen et al., 2018*), commonly by two glycine-binding GluN1-subunits (encoded by *GRIN1* gene) combined with two glutamate- (*GRIN2A-D*) or glycine-binding (*GRIN3A-B*) subunits. GluN1 subunits are an obligatory subunit of all channel types and are therefore found throughout the brain and during all stages of life, whereas GluN2B- and GluN2D-subunits are particularly abundant during embryonic stages (*Paoletti et al., 2013*; *Hansen et al., 2018*; *Xu and Luo, 2018*). Conversely, GluN2A and GluN2C-subunits increase after birth, and GluN2A further replaces GluN2B during maturation of excitatory synapses (*Paoletti et al., 2013*; *Lau and Zukin, 2007*). This variety of subunits gives rise to a large combinatorial diversity of channel subtypes in neurons (*Paoletti et al., 2013*; *Lau and Zukin, 2007*) (but also glia [*Verkhratsky and*

*Kirchhoff, 2007*]) with each channel-type displaying unique biophysical and pharmacological properties, and differential patterns (and timing) of expression (*Paoletti et al., 2013*; *Hansen et al., 2018*). Hence, owing to their essential roles in the brain, dysfunctional GluNRs are highly associated with a myriad of diseases of the brain (*XiangWei et al., 2018*; *Yuan et al., 2015*).

Since 2010, thousands of different *GRIN* variants have been discovered in pediatric patients (*Xu and Luo, 2018*; *XiangWei et al., 2018*; *Myers et al., 2019*), with the majority of mutations predominantly concentrated within the *GRIN2A* and $-2B$ genes (46% and 38%, respectively) (*XiangWei et al., 2018*; *Myers et al., 2019*; *García-Recio et al., 2021*). To date, a small fraction of mutations have been functionally characterized, notably <15% reported for *GRIN2B* (*Xu and Luo, 2018*; *XiangWei et al., 2018*; *Tang et al., 2020*; *Platzer et al., 2017*) (see full list of mutation in *Supplementary files 1* and *2*). Pathogenic variants in *GRIN* genes cause severe encephalopathies, with complex and overlapping clinical pictures involving intellectual disabilities (ID), neurodevelopmental delays (DD), autism spectrum disorders (ASD), movement and language disorders, schizophrenia, seizures, epilepsy and more (*XiangWei et al., 2018*; *Yuan et al., 2015*; *Myers et al., 2019*). Currently, there are no specific treatments for different *GRIN*-diseases, rather patients receive supportive care and/or are specifically treated for the different co-morbidities of the disease (e.g. anti-seizure medication). Nevertheless, in the case of Gain-of-Function (GoF) mutations, there are several exploratory treatments with non-specific GluNR-blockers as primary candidates, notably the FDA-approved drug memantine (*Pierson et al., 2014*; *Strehlow et al., 2019*; *Amador et al., 2020*; *Lipton, 2006*). Loss-of-Function (LoF) mutations are harder to treat, as there are few channel openers, and even fewer subunit-selective (e.g. *Tang et al., 2020*; *Mony et al., 2009*; *Costa et al., 2010*; *Perszyk et al., 2018*; *Warikoo et al., 2018*). Moreover, none are FDA-approved (*Wilkinson and Sanacora, 2019*; *Silva et al., 2019*). A recent report suggests an alternative approach consisting of the use of L-serine for enhancing channel activity (*Soto et al., 2019*). L-serine is converted to D-serine; the co-agonist of the GluN1 subunit (*Neame et al., 2019*). This supplementation was shown to improve the condition of children with a mild *GRIN2B* LoF mutation (<sevenfold reduction in glutamate affinity) (*Soto et al., 2019*) and is currently under clinical trials (*de Déu, 2020*). However, it is worth noting that before treatments can be suggested (even if experimental), it is essential to understand the effects of the mutations on channel function (e.g. GoF *vs* LoF) because of the potential to worsen symptoms and provoke devastating effects (i.e. excessive channel activation, unwarranted cellular excitability, cytotoxicity, or neurodegeneration [*Hardingham et al., 2002*; *Yan et al., 2020*; *Choi, 1987*; *Lipton and Rosenberg, 1994*]) if, for example, an 'opener' is provided to treat a GoF mutation. These highlight the importance in curating and functionally characterizing each mutation before treatment(s) can be formulated (*García-Recio et al., 2021*).

Here, we provide functional characterization of two *de novo* GRIN2B mutations, occurring at the same residue (p.G689) in two pediatric patients; an Israeli patient with a G689C mutation and a Danish patient with G689S. The two patients have been diagnosed with a severe *GRIN2B*-encephalopathy, both showing similar—but also diverging—clinical symptoms (*Table 1*). We surveyed the literature and found that the G689C is a novel mutation, whereas the G689S variant has been noted in two previous patients though, to the best of our knowledge, characterization of the G689S-variant has not been reported (*Platzer et al., 2017*; *Bosch et al., 2016*).

We simulated the structures of the LBD with G689C or G689S and find the two variants to destabilize the glutamate-binding pocket; suggesting them to prompt LoF. Indeed, scrutiny of channel function showed that both variants show very strong reductions in glutamate potency, with G689S presenting a more drastic shift in $EC_{50}$ (G689C- ~1000-fold and G689S- ~2000-fold reduction). We also find that while G689C express poorly at membranes of cells (and therefore yields lower current amplitudes), G689S does not. The latter stands in contrast to the common notion that reduced glutamate potency correlates with reduced surface levels of the receptors (*Swanger et al., 2016*; *She et al., 2012*). Yet, despite differences in expression levels of the variants, both enhance the expression of GluN2B*wt*-subunits and exert a very strong dominant-negative effect on receptor function. In primary neurons, these features translate into reduced synaptic NMDAR-dependent currents. We go on to explore channel potentiators (e.g. spermine), however, find that both variants fail to respond under physiological conditions. Scrutiny of the reasons behind this observation led us to discover that G689 is an essential residue in the elusive proton-sensor of the GluN2B subunit. Together, we describe two exceptional mutations in two patients exhibiting a diverging clinical picture. We demonstrate the two first cases showing LBD-mutations that exert a *bona-fide* dominant negative

**Table 1.** clinical proband of patients suffering from De novo GRIN2B mutations.

| Mutation | *GRIN2B* p.Gly689Ser | *GRIN2B* p.Gly689Cys |
|---|---|---|
| Genotype (type) | c.2065G>A, De Novo, heterozygous (WES; performed at ~1 year of age) | c.2065G>T, De Novo, heterozygous (WES; performed at 8 months of age) |
| CMA | N/A | Normal |
| Age | 6 | 3.5 |
| EEG | Initial EEG (before 1 y): epileptiform discharges 1.5- year -old: diffuse mix of beta activity, sometimes sharp waves centrally in the midline; sleep: recurrent trains of sharp waves / polyspikes and slow waves with a very high amplitude, as well as trains of fast activity (20 Hz) occipito-post-temporal, sometimes with a diffuse spread (correlated with myoclonic seizures) Latest EEG: May 2019: Spikes / Sharp-waves in the left temporo/central region | Normal with no epileptic behavior |
| Seizures | Myoclonic seizures + epileptic spasms | No epileptic activity |
| MRI | Normal (at 1 year of age) | Asymmetry in left ventricle (up to 11 mm; prenatally) Normal (at 1.5 year of age) |
| DD | Severe, crawl for short distance | Severe, can crawl, walk with aid |
| ID | Severe – nonverbal | Severe - speaks 10 words, understands simple commands |
| Strabismus | No | Yes |
| Gastrointestinal symptoms | N/A | Constipation (until two years old), reflux |
| Additional observations | Dyskinetic movements, hypertonia | Dyskinetic movements (resolved at age of 2), hypotonia, hyperflexible, no dysmorphism |
| Current medication | Piracetam and serine Previously treated with CBD and valproic acid | N/A |
| VUS | SLC6A8, CACNA1A | N/A |
| Collaboration | Danish Epilepsy Centre | The Genetics institute -Rambam |

CMA – chromosomal microarray analysis; DD -developmental delay; ID – intellectual disability; VUS - Variant of uncertain significance; N/A – not available.

effect in oocytes and primary cultured neurons. Lastly, we estimate the effect of more than 5200 mutations on the stability of the LBD and these data help to explain why most LBD mutations instigate Loss-of-Function and further suggest that substitutions to glycine or serine are the most damaging to the LBD.

## Results

### Two different variants occurring at the same residue in the ligand-binding domain of GluN2B are associated with neurodevelopmental disorders

We have identified a toddler with a previously unknown, *de novo*, heterozygous missense mutation in the *GRIN2B* gene. The mutation consists of a single base-pair substitution in the gene (c.2065G>T), thereby changing a highly conserved glycine into a cysteine residue at position 689 (G689C, *Figure 1a*). This residue is located deep within the ligand-binding-domain (LBD) of the receptor (*Figure 1b*), adjacent to *all* residues directly involved in agonist-binding (*Figure 1—figure supplement 1a*; *Traynelis et al., 2010*). We found another pediatric patient in Denmark carrying an analogous mutation, specifically G689S. We combed the literature and found that although the G689S mutation has been previously reported, it has not been characterized (*Platzer et al., 2017*; *Bosch et al., 2016*). Of note, the incidence of two different variants affecting the same residue within *GRIN2B* is relatively low (<15%, Extended data 1), for instance the N615K and N615I variants (*Xu and Luo, 2018*; *XiangWei et al., 2018*; *Fedele et al., 2018*; *Amin et al., 2018*). Both *GRIN2B*-G689x (x; cysteine or serine)-patients exhibit a similar clinical picture of severe DD, ID, and dyskinetic movements. However, the toddler with the G689S variant also presents myoclonic seizures and

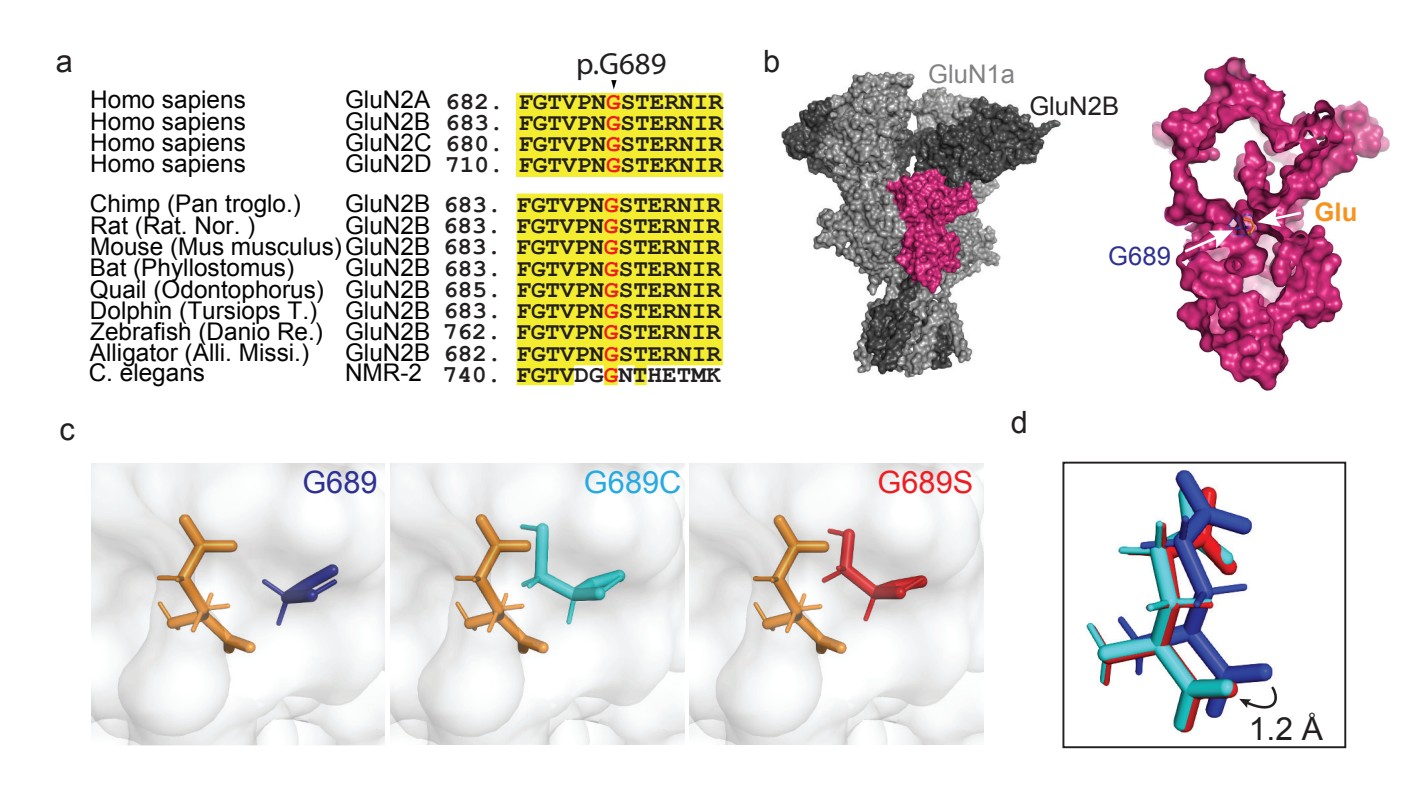

**Figure 1.** *De novo* mutations occurring at a highly conserved glycine residue in the human GluN2B subunit. (a) Sequence alignment, showing strong homology and conservation of Glycine residue at position 689 (black arrowhead) among different GluN2 subunits (A–D) and different species. (b) Crystal structure of GluN1a-2B (PDB 4PE5 [*Karakas and Furukawa, 2014*]). Left image- light and dark gray depicts GluN1a and GluN2B subunits respectively; GluN2B LBD is highlighted in pink. Inset- Space filling of the GluN2B's LBD showing the deep location of Gly689 residue (blue), positioned adjacent to the bound glutamate ligand (orange). (c) Prediction of *de novo* mutations occurring at p.689; left image- p.Gly689 (dark blue); center image- p.Gly689Cys (cyan) and right- p.Gly689Ser (red). Note that the side chains of the mutations point toward the glutamate ligand. This apposition required structural shifting of the glutamate ligand to prevent clashes with the original position, as shown in (d). (d) Simulation of the shift (1.2 Å) in the position of the glutamate ligand, resulting from mutagenesis of G689. Original position (as in PDB 4PE5) is shown in dark blue (representing the presence of the original G689). Cyan and red glutamates correspond to the newly positioned glutamate ligands obtained by the simulations when G689 was replaced by C or S, respectively.

The online version of this article includes the following figure supplement(s) for figure 1:

**Figure supplement 1.** G689 lies adjacent to essential residues involved in glutamate binding.

**Figure supplement 2.** G689C and G689S mutations block possible water exit tunnels.

**Figure supplement 3.** GluN2B LBD unbound cysteine residues.

**Figure supplement 4.** Mutations of G689 yield destabilized LBD.

spasms (*Table 1*), even though seizures are more associated with mutations in *GRIN2A* (*XiangWei et al., 2018*; *Myers et al., 2019*). Thus, despite the overlapping clinical picture, the differences between the conditions of the patients—even if slight—suggest that the variants should differently affect receptor function.

## Variants exhibit a destabilized LBD and glutamate-binding pocket

To try to understand the effect of the mutations before functional assessments, we initially simulated the structures of the LBD of GluN2B by replacing G689 by a cysteine or a serine residue in the reported GluN1a/GluN2B structure (PDB 4PE5 [*Karakas and Furukawa, 2014*]). Briefly, we isolated the LBD of one of the GluN2B monomers from the tetrameric structure, introduced the mutations, followed by protein preparation and energy minimization (see Materials and methods). Although the general structure of the LBD appears to remain intact, the simulations predict the side chains of the two mutations to point toward the ligand and, thereby, to occupy a larger volume in the glutamate

binding pocket, as opposed to the original glycine which points away (*Figure 1c*, cysteine-cyan, serine-red). This apposition required shifting of the bound glutamate (by ~1 Å) to prevent clashes with the original position (*Figure 1d*). This is accompanied by similar shifts in all residues involved in binding and coordination of glutamate (*Figure 1—figure supplement 1b*). Together, these strongly suggest destabilization of the glutamate-binding pocket by steric interference (*Figure 1d*). The simulations also show that the side chains of the variants completely block water-accessible entry sites found in the glutamate-bound and closed state of the LBD (*Figure 1—figure supplement 2*), and these are implicated in modulating deactivation kinetics (*Swanger et al., 2016*; *Wells et al., 2018*). Lastly, the cysteine in the G689C variant remains sufficiently distant from other naturally occurring cysteines in the LBD and should therefore remain unbound (*Figure 1—figure supplement 3*). Nevertheless, the latter does not preclude the possibility that the cysteine may interact with additional cysteines during translation and folding of the subunit, in which case should negatively impact the variant's expression levels (*Feige et al., 2018*). If correct, then the G689S variant should have less of an effect on expression levels of mutant channels. Further analysis of protein stability (ΔΔG; by employing Mutation Cutoff Scanning Matrix [*Pires et al., 2014*]) shows that, whereas all possible mutations of the G689 residue yield a negative ΔΔG, both mutations are among the substitutions that mostly impact the stability of the LBD (*Figure 1—figure supplement 4* and see below). Together, the simulations and assessment of protein stability suggest that both variants interfere with glutamate-binding and likely instigate LoF. Our simulations also point toward minor effects on deactivations kinetics and potential impact on expression levels of G689C.

## The G689C and G689S variants show drastically reduced glutamate potency

We turned to functionally characterize the effects of the mutations on glutamate potency (EC$_{50}$). Therefore, we co-expressed the human GluN1A-*wt* (hGluN1a) subunit with three different human GluN2B variants, namely hGluN2B*wt*, 2B-G689C, or 2B-G689S, in *Xenopus* oocytes and assessed receptor function by two-electrode voltage clamp (TEVC). Briefly, oocytes were held at −60 mV and perfused with Mg$^{2+}$-free (to avoid voltage-dependent block [*Hansen et al., 2018*]) and Ca$^{2+}$-free solutions (to avoid Ca$^{2+}$-activated chloride currents in oocytes [*Mony et al., 2011*; *Berlin et al., 2016*]), with incrementing concentrations of glutamate (as in *Berlin et al., 2016*, see Materials and methods). We initially conducted experiments using glutamate concentrations suitable for the hGluN2B*wt* receptors (e.g. *Paoletti et al., 2013*) and find the *wt*-receptors to respond to very low glutamate concentrations (here, 0.2 μM), with micromolar glutamate potency (EC$_{50}$ = 1.4 ± 0.04 μM, n = 43; *Figure 2a* and *Table 2*; *Paoletti et al., 2013*; *Traynelis et al., 2010*). Under these conditions, however, we could barely detect glutamate-dependent currents from oocytes injected with the hGluN2B-G689C mutant, and even less so in oocytes expressing the G689S variant (*Figure 2—figure supplement 1a and b*, respectively). Glutamate-dependent currents appeared only at ~mM concentrations (*Figure 2b,c* and *Figure 2—figure supplement 1a,b*; 2 mM). Importantly, glutamate-currents were not observed in uninjected oocytes (*Figure 2—figure supplement 1c*), unless when glutamate concentrations exceeded 10 mM (*Figure 2—figure supplement 1c,d*). We therefore did not apply glutamate concentrations past 10 mM when assessing glutamate efficacy of the variants. On a side note, we find others to use 10 mM glutamate as an upper limit when performing glutamate dose-responses, although the authors do not provide the rationale for this choice (*Lemke, 2016*). Regardless, under these conditions, whereas the hGluN2B-*wt* channels were fully saturated (*Figure 2b* and *Figure 2—figure supplement 1e*), the variants display >1000-fold reduction in EC$_{50}$, with G689S presenting a more severe phenotype (glutamate EC$_{50}$: hGluN2B-G689C = 1.54 ± 0.14 mM [1100-fold], n=31; hGluN2B-G689S = 2.56 ± 0.14 mM [1814-fold], n=23; *Figure 2a,b*, *Table 2*). To the best of our knowledge, this is the largest reduction in affinity observed for GluN2B-mutants (*XiangWei et al., 2018*; *Platzer et al., 2017*; *Swanger et al., 2016*). Thus, our observations confirm the predicted LoF and complement previous studies showing that most LBD mutations lead to LoF (*Myers et al., 2019*; *Swanger et al., 2016*) (see *Supplementary file 1*). We also observed that hGluN2B-G689C-containing channels—but not GluN2B-G689S—exhibit significantly smaller currents (I$_{max}$~40%) than those of hGluN2B*wt*-containing receptors (*Figure 2c*) and lower glycine affinity (*Figure 2—figure supplement 2a,b*). However, neither present alteration in Mg$^{+2}$-sensitivity (*Figure 2—figure supplement 2c,d*, summarized in *Table 2*). Together, the G689C

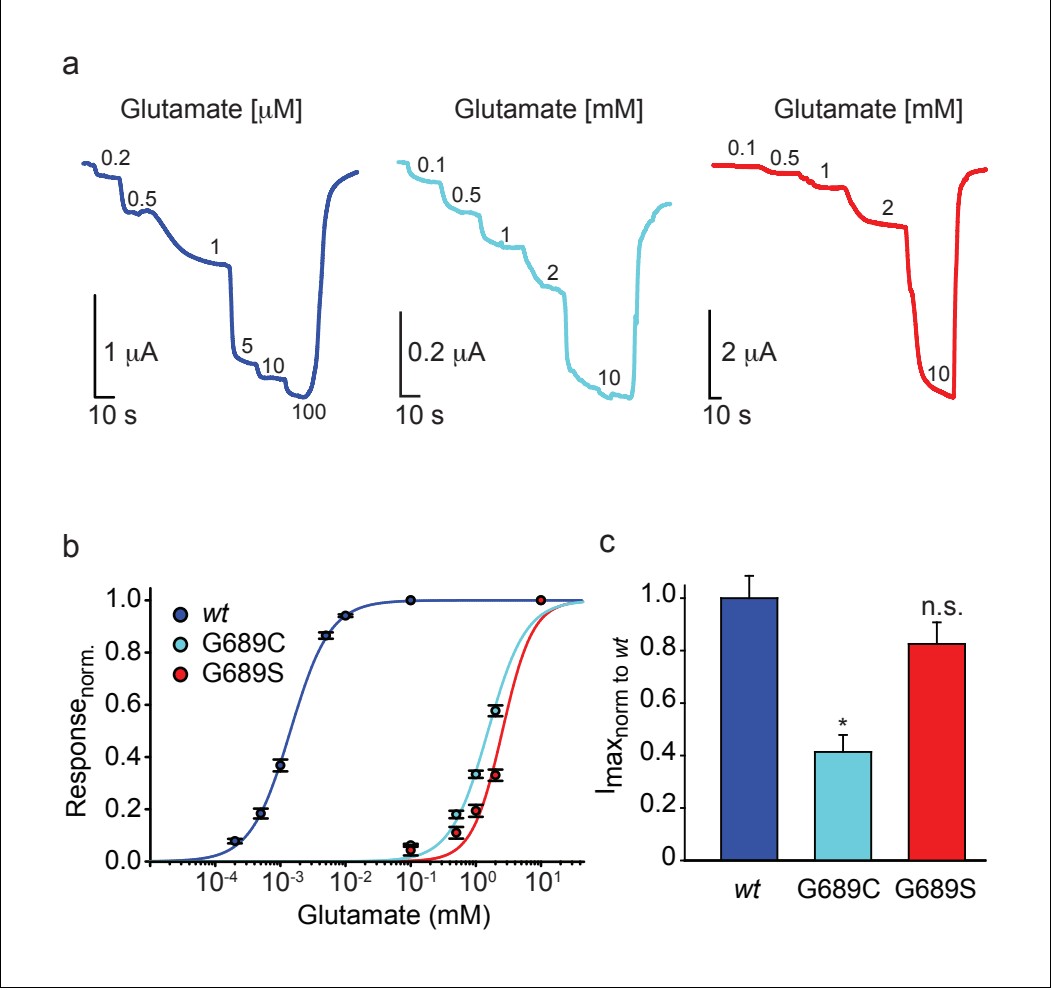

**Figure 2.** GluN2B LBD mutations drastically reduce glutamate potency. (a) Representative glutamate-dependent currents recorded from *Xenopus* oocytes co-expressing hGluN1a with GluN2B-*wt* (blue); GluN2B-G689C (cyan) or GluN2B-G689S (red). Glutamate concentrations are marked next to current steps (note the differences in units, namely µM and mM); summarized in (b). (c) Summary of the normalized maximal current ($I_{max}$) of the different GluN2B-subunits, showing the significantly smaller currents of G689C, but not G689S, mutant. For (b), *wt*- 43 cells; G689C- 31 cells; G689S- 23 cells, collected form two to three independent experiments; (c), *wt*- 22 cells; G689C-30 cells; G689S- 21 cells, collected form one to two independent experiments.

The online version of this article includes the following figure supplement(s) for figure 2:

**Figure supplement 1.** Assessment of the effect of low to very high glutamate concentrations on uninjected oocytes and G689C/S-injected oocytes.

**Figure supplement 2.** Variants' effect on Glycine and $Mg^{2+}$-sensitivity.

and G689S variants induce a severe LoF, although to different extents. These dissimilarities could reflect the differences observed between the clinical phenotypes (*Tables 1* and *2*).

## GluN2B variants show reduced surface expression in HEK293 cells

The reduced current-amplitudes of G689C-containing channels suggests differences in expression levels of the receptors. We therefore addressed expression levels by β-lactamase activity. Briefly, we tagged the extracellular amino termini of the various hGluN2B-subunits with β-lactamase and incubated transfected cells with the cell-impermeable β-lactamase substrate (nitrocefin) and measured expression by extracting the slopes from continuous absorption measurements obtained by a plate reader (see Materials and methods and [*Swanger et al., 2016*]). The results obtained from multiple independent experiments show that the 2B-G689C variant expresses the least at membrane of

**Table 2.** Summary of pharmacological profiling for hGluN2B-G689C and hGlun2B-G689S.

| Variant | Glutamate EC$_{50}$ (n) | I$_{max}$ Norm. to wt (n) | Glycine EC$_{50}$ (n) | Mg$^{+2}$ IC$_{50}$ (−60 mV) (n) | 10–90% inhibition rate (n) | $\tau_{off}$ (n) | Proton IC$_{50}$ (n) |
|---|---|---|---|---|---|---|---|
| hGluN2B-wt | 1.4 ± 0.04 μM; (43) 0.8 ± 0.01 μM; (6, HEK293T cells) | 1 ± 0.08; (22) | 0.23 ± 0.03 μM; (33) | 31 ± 3.7 μM; (33) | 8554 ± 827 ms; (24) | 4590 ± 603 ms; (24) | 7.26 ± 0.02; (15) |
| hGluN2B-G689C | 1.54 ± 0.14* mM; (31) | 0.41 ± 0.06*; (30) | 0.4 ± 0.05** μM; (33) | 34.5 ± 3.8 μM; (17) | 9604 ± 547 ms; (29) | 4950 ± 456 ms; (29) | 7.04 ± 0.01*; (19) |
| hGluN2B-G689S | 2.56 ± 0.40* mM; (23) 2.2 ± 0.39 mM *; (20, HEK293T cells) | 0.82 ± 0.08; (21) | 0.31 ± 0.03 μM; (15) | 40 ± 2.1 μM; (23) | 18904 ± 1375* ms; (26) | 7907 ± 658* ms; (26) | 6.96 ± 0.03*; (21) |

mammalian cells (HEK293T cells), at levels corresponding to ~45% of GluN2Bwt-containing receptors (*Figure 3a,b*), without any apparent differences in channel open probability (P$_o$) (*Figure 3—figure supplement 1a–c*). The 2B-G689S-variant, on the other hand, expressed as well as *wt*-receptors, in spite of the small reduction in P$_o$ (*Figure 3b*, red and *Figure 3—figure supplement 1a–c*, red). These results provide the reason behind the lower current amplitudes obtain for 2B-G689C (~40%) (see *Figure 2c*).

Owing to the heterozygosity nature of the disease, we next wondered whether the mutant subunits could also affect the expression of GluNBwt-subunits or mimic cases of haploinsufficiency, in which case there would simply be less channels (here less responsive channels) at the membrane (*García-Recio et al., 2021*). We have therefore co-expressed constant amounts of GluN1a and GluN2Bwt-β-lac with incrementing concentrations of the 2B-G689C or 2B-G689S variant (spanning an order of magnitude, see Materials and methods). Surprisingly, both variants significantly improved the surface expression of receptors expressing the GluN2Bwt-β-lac subunit (*Figure 3c*). These increases were not seen when co-expressing another control plasmid encoding for a non-

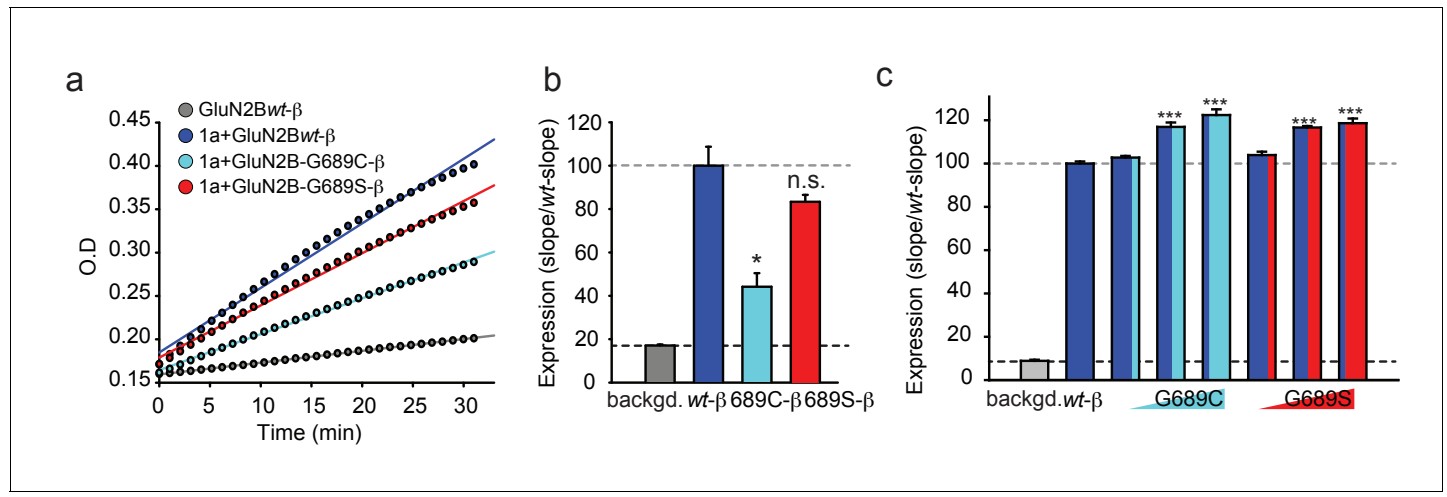

**Figure 3.** The G689C variant is poorly expressed at membranes of cells, but both variants enhance expression of GluN2Bwt. (a) Representative plot of nitrocefin absorbance (O.D) over time (min) in HEK293T cells expressing GluN2B-β-lac (gray), hGluN1a-GluN2B-wt-β-lac (blue), hGluN1a-GluN2B-G689C-β-lac (cyan), hGluN1a-GluN2B-G689S-β-lac (red). Background signal (backgd.; gray) was collected from cells expressing GluN2Bwt-β-lac only (wt-β, 1 μg DNA) without GluN1a (thereby not transported to membrane). (b) Summary of expression of the various channel types from three to five independent experiments. Colors are as depicted in (a). Results show that G689C expresses at ~44% compared to the expression of 2B-wt or G689S subunits. (c) Summary of the expression of GluN2Bwt-β-lac (wt-β, 1 μg DNA), co-transfected with 1 μg GluN1a, when also co-transfected with increasing DNA amounts (0.1, 0.5, and 1 μg) of the G689C (cyan) or G689S variants (N = two independent experiments). *, p<0.05; **, p<0.01; ***, p<0.001; n.s. non-significant. Statistics show comparison to the wt-β group.

The online version of this article includes the following figure supplement(s) for figure 3:

**Figure supplement 1.** hGluN2B-G689S, but not hGluN2B-G689C, shows reduced apparent open probability (P$_o$).

NMDAR-related channel subunit (Kv4.2) (*Figure 3—figure supplement 1d*). In fact, incrementing DNA amounts of Kv4.2 reduced the surface levels of the GluN2B*wt*-β-lac subunit, likely by competing for translation, as shown earlier for other channels and proteins (*Berlin et al., 2020*). Thus, increases in surface levels of GluN2B*wt*-β-lac (and also of the mandatory GluN1a-subunit) suggests that the variants multimerize with the GluN2B*wt*-subunits to form tri-heteromeric channels ('mixed channels') with a 2:1:1 stoichiometry, namely two copies of GluN1a-*wt* coupled with one copy of GluN1B*wt* and another copy of the GluN2B-mutant.

## The two variant exert a dominant-negative effect over GluN2B-wt in *Xenopus* oocytes

We were next interested in examining the functional outcome of the collection of data showing that G689C-containing channels—but not G689S—express poorly at membranes of mammalian cells (*Figures 2* and *3*), along the observations that both variants increase the membrane levels of the GluN2B*wt*-receptors (*Figure 3*). To do so, we co-expressed hGluN1a with different mixtures of hGluN2B-*wt* and hGluN2B-G689C or -G689S mRNAs in *Xenopus* oocytes (i.e. mRNA titrations [*Berlin et al., 2020*; *Berlin et al., 2011*; *Yakubovich et al., 2015*; *Peleg et al., 2002*; *Katz et al., 2021*]) and assessed glutamate dose-responses (see Materials and methods). When large mRNA amounts of hGluN2B-*wt* were co-expressed with very low amounts of hGluN2B-G689C (mRNA ratio: ~16:1, denoted wildtype-high; $wt_H$), we recorded large currents that readily responded to glutamate (*Figure 4a,b*; $wt_H$). These features are indicative of the predominant expression of the hGlu2B*wt*-containing receptors. However, the dose-response curve for this group could not be fitted for a single population, rather was best fitted bimodally (Materials and methods and *Ben-Chaim et al., 2003*). This strongly suggested *two* receptor populations with differing affinities for glutamate, namely high and low (*Figure 4c*; $wt_H$). We tested several different adjustments to the fits to ensure that we did not over-represent the data for the G689C group (see Materials and methods, *Equations 2 and 3*) and, indeed, obtained several different low and high apparent KDs ($KD^L$ and $KD^H$). In the $wt_H$ group co-expressing 2B-G689C, no matter the fitting procedure, the different $KD^H$ obtained by the various fits (0.44 or 0.21 μM, *Figure 4—figure supplement 1a* and *Table 3*) were on par with those obtained for the single *wt* population (*wt*- $EC_{50}$ = 1.4 μM, see *Figure 2*); however, $KD^L$ did not match the $EC_{50}$ obtained solely for G689C receptors (1.54 mM vs. 0.37 or 0.15 mM, *Table 3*). We also fitted the same curves after these have been normalized to the responses obtained by 100 μM glutamate (assuming saturation of all *wt* receptors). This handling left $KD^H$ constant (as above), but only slightly increased $KD^L$ (*Figure 4—figure supplement 1a*, blue traces and *Table 3*). Thus, the data demonstrate that expression of GluN2B*wt* mRNA with very low amounts of GluN2B-G689C yields two receptor-populations: one closely resembling full *wt* receptors (i.e. GluN1a-*wt* and GluN2B*wt*), and another population of mixed-channels exhibiting three orders of magnitude lower $EC_{50}$ than *wt* receptors (from 0.44 μM to 0.37 mM). We proceeded to examine $wt_E$ (*E*qual amounts of mRNA between *wt* and variant) and $wt_L$ (*L*ow amounts of *wt* mRNA), co-expressed with G689C. We first note that both groups yield similar $I_{max}$ as the $wt_H$ group (*Figure 4a*, summarized in b). This is particularly surprising for the $wt_L$ group containing 16-fold more mRNA of the G689C variant than GluN2B*wt*. In this instance, we expected the currents to be dominated by G689C-containing channels and thereby of lower expression and current-amplitudes (see *Figure 2*). Interestingly, these results are consistent with our expression assays showing that G689C does not interfere, rather promotes the expression of GluN2B*wt*-containing receptors (see *Figure 3* and *Figure 3—figure supplement 1d*). Scrutiny of the glutamate dose-response curves for $wt_E$ and $wt_L$ showed pronouncedly right-shifted curves (*Figure 4c*, purple and turquoise curves). We find that, although all curves were best fitted by bimodally as $wt_H$, the proportion of the $KD^H$ part of the curve was dramatically decreased, expectedly showing that the contribution of the affinity of GluN2B*wt* subunits is diminished (relative responses to 100 μM glutamate: 60% for $wt_H$; 16% for $wt_E$; 5% for $wt_L$). Second, the $KD^L$ values were also reduced dramatically, both in proportion (increase) and in values corresponding better to those obtained for when 2B-G689C channels were assessed alone (*Tables 2* and *3*). Together, these observations demonstrate that the GluN2B*wt* subunits are readily expressed in all three groups. These are attested by: (1) the size of current, (2) by the high-affinity population in dose-response curves and further supported (3) by expression assays. However, even at low expression levels of G689C variant ($wt_H$, $wt_E$), the apparent affinities for glutamate are persistently diminished. Similar results were obtained when GluN2B-G689S was co-expressed with GluN2B*wt*

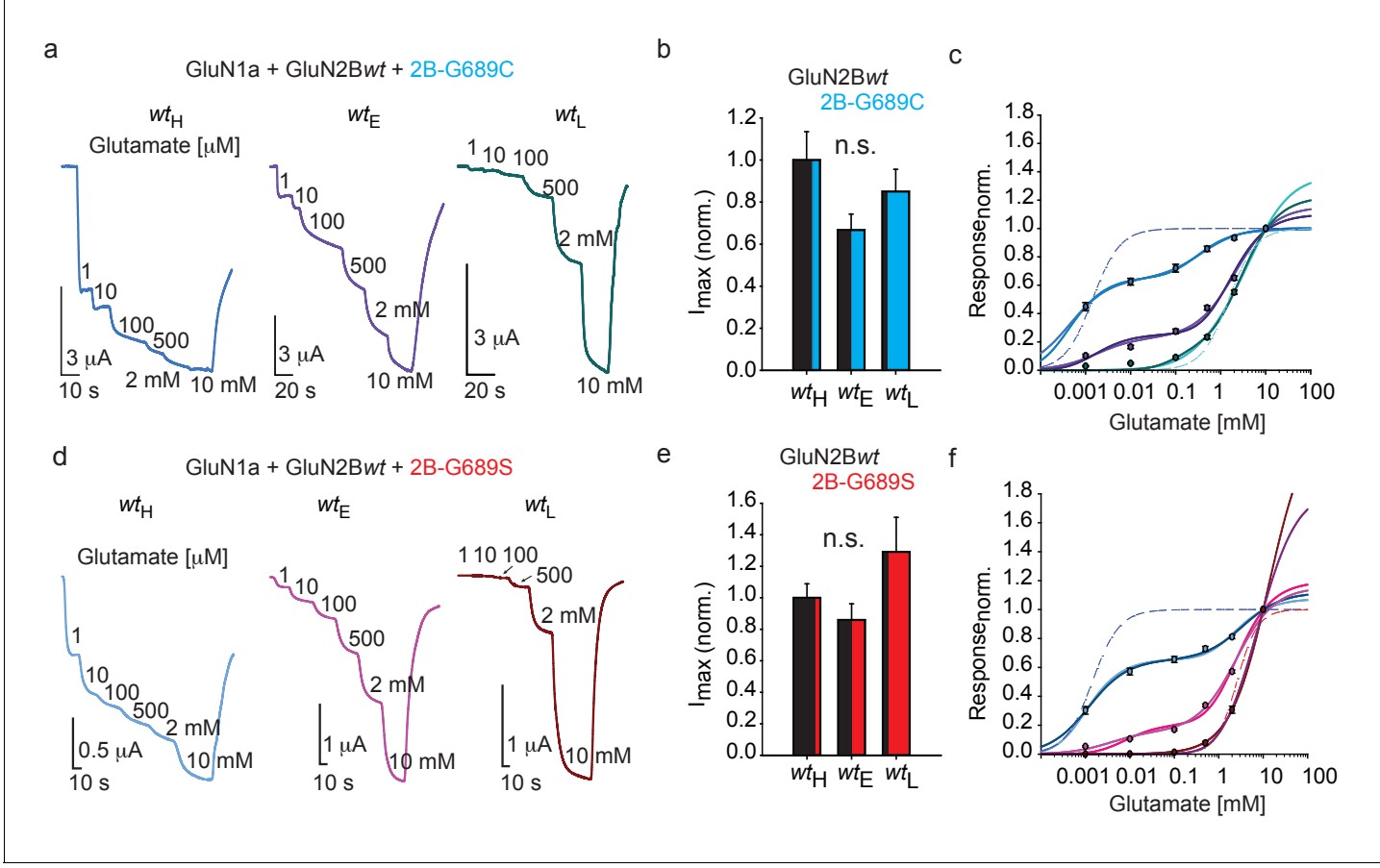

**Figure 4.** GluN2B LBD variants exert a dominant-negative effect over GluN2B-*wt* subunit in *Xenopus* oocytes. (**a** and **d**) Representative traces from oocytes co-expressing hGluN1a, hGluN2B-*wt* and GluN2B-G689C (**a**) or G689S (**d**) at different mRNA ratios in response to incrementing glutamate concentrations (indicated next to trace, in µM unless mentioned otherwise). Oocytes in which we co-injected mRNA of GluN2B*wt* 16-fold over mRNA of variant belong to wildtype-High group (*wt*$_H$). Oocytes co-injected with equal mRNA amounts belong to *wt*-Equal group (*wt*$_E$) and, inversely, oocytes co-injected with 16-fold more mRNA of variant over GluN2B*wt*, belong to the *wt*-Low group (*wt*$_L$). (**b** and **e**) Summary of the normalized maximal current ($I_{max}$) obtained by 5 mM glutamate and 100 µM glycine for the different treatments (*wt*$_H$, *wt*$_E$ and *wt*$_L$) when G689C (**b**) or G689S (**e**) were co-injected with the GluN2B*wt*-subunit. N = 1–2 experiments, n = 10–19 oocytes. (**c** and **f**) Glutamate dose-response curves for the three different treatments with G689C (**c**), namely *wt*$_H$, *wt*$_E$ and *wt*$_L$ groups fitted by different bimodal curves in blue-, purple-, and cyan-shades, respectively (values are shown in *Table 3*) and for G689S (**f**) by light blue-, pink- and cyan shades, respectively. Dashed plots (blue and cyan in **c**; blue and pink in **f**) show dose-response curves for di-heteromeric channels, hGlun1a-hGluN2B-*wt* and hGlun1a-hGluN2B-G689C or G689S, respectively (as shown in *Figure 2b*). For (**c**), *wt*$_H$- 18 cells; *wt*$_E$ - 26 cells; *wt*$_L$- 19 cells, collected form two to three independent experiments; (**f**), *wt*$_H$- 14 cells; *wt*$_E$ - 11 cells; *wt*$_L$- 12 cells, collected form one to two independent experiments.

The online version of this article includes the following figure supplement(s) for figure 4:

**Figure supplement 1.** Re-normalization of dose-response curves to 100 µM glutamate persistently shows dominant-negative effect of variants at *wt*$_H$ group.

(*Figure 4d–f*, *Figure 4—figure supplement 1* and *Table 3*). Collectively, our functional results complement our expression assays and suggest that both variants readily co-assemble with the *wt*-subunits to form tri-heteromeric channels at membrane of cells, without affecting current size and expression of the channels. This effect is unique as similar reports suggest that mixed channels should yield reduced currents (e.g. *Lemke, 2016*; *Endele et al., 2010*). Importantly, both variants strongly reduce the glutamate affinities of the tri-heteromeric channels (approaching the affinities of the variants when these are expressed alone); demonstrating their strong dominant negative effect.

Dominant negative effects in *GRINs* is quite uncommon. In support, we find only a handful (three) of reports that explicitly note a dominant-negative effect for mixed channels for three *GRIN* variants (*GRIN1 Lemke, 2016*, *GRIN2A* (*Endele et al., 2010*) and GRIN2B [*Li et al., 2019*]). In two reports (*Lemke, 2016*; *Endele et al., 2010*), the authors interpret reduced current amplitudes (~50%) as an

**Table 3.** Glutamate dose-response fitted parameters.

| | $EC_{50}$ Two site saturation (*Equation 2*) $KD^H$ (top row), $KD^L$ (bottom row) | $EC_{50}$ Two site saturation (*Equation 3*) $KD^H$ (top row), $KD^L$ (bottom row) | N |
|---|---|---|---|
| $wt_H$ (16:1 G689C) | 440 nM ± 57 nm<br>366.5 µM ± 74.4 µM | 212 nM ± 32 nM<br>148 µM ± 37 µM | 18 |
| $wt_H$ (16:1 G689C) – normalized to 100 µm glutamate | 430 nM ± 76 nm<br>370 µM ± 93.3 µM | 212 nM ± 25 nM<br>154 µM ± 27 µM | 18 |
| $wt_E$ (1:1 G689C) | 1.725 µM ± 1.541 µM<br>2.128 mM ± 728 µM | 812 nM ± 818 nM<br>778 µM ± 320 µM | 26 |
| $wt_L$ (1:16 G689C) | 646 µM ± 3.716 mM<br>5.492 mM ± 33.48 mM | 65 µM ± 79 uM<br>1.426 mM ± 648 uM | 19 |
| $wt_H$ (16:1 G689S) | 11.85 µM ± 154 nM<br>3.460 mM ± 1.233 mM | 490 nM ± 90 nM<br>1.127 mM ± 487 µM | 14 |
| $wt_H$ (16:1 G689S) – normalized to 100 µm glutamate | 1.15 nM ± 164 nm<br>3.39 mM ± 1.25 mM | 481 nM ± 94 nM<br>1.12 mM ± 491 µM | 14 |
| $wt_E$ (1:1 G689S) | 3.503 µM ± 5.162 µM<br>2.460 mM ± 1.081 mM | 3.792 µM ± 3.946 µM<br>1.208 mM ± 406 µM | 11 |
| $wt_L$ (1:16 G689S) | 13.34 mM ± 4.68 nM<br>13.34 mM ± 2.53 nM | 1.593 mM ± 78.62 mM<br>6.261 mM ± 1498 mM | 11 |

indication for dominant negative effect, whereas the third report indicates that a single copy of GluN2B-N616K produces a dominant reduction in $Mg^{2+}$-block similar to channels including two copies of the variant (*Li et al., 2019*). However, most reports examining other *GRIN* mutations do not describe dominance. For instance, a recent report examining eight different *GRIN* variants (M2-pore mutations) shows that mixed-channels exhibit very mild reduction in $Mg^{2+}$ $IC_{50}$, with values corresponding to values of the *wt* channels (*Li et al., 2019*). Another report examining mixed-channels containing GluN2A*wt* and 2A-P552R (*Ogden et al., 2017*) shows that, whereas 2A-P552R significantly alters stability of the pore when it is found in two copies per channel, it fails to do so when mixed with GluN2A*wt*. Very similar observations are reported for mixed channels bearing 2B*wt* and the GluN2B-E413G variant in which the $EC_{50}$ is not dominated by the low-affinity subunit (*Swanger et al., 2016*) (and see Discussion). Thus, while a dominant negative effect is somewhat intuitive—as all LBDs of NMDARs need to be liganded for full channel opening in which case the *weakest* subunit would be the limiting factor (*Berlin et al., 2016*; *Wilding et al., 2014*; *Kussius and Popescu, 2009*)— it is not commonly reported for *GRIN* mutations, especially not for LBD mutations in *GRIN2B*.

Together, our observations suggest a dual effect by the variants. First, mutant receptors bearing two copies of the variants are (likely) completely non-functional physiologically. This can be potentially related to cases of haploinsufficiency (with 50% the amount of the protein [*García-Recio et al., 2021*; *Santos-Gómez et al., 2021*; *Shin et al., 2020*]); however, the normal expression of the G689S variant challenges this categorization. Third, the variants instigate a strong dominant-negative effect over glutamate potency when combined with 2B*wt* subunits (see Discussion).

## Spermine weakly potentiates GluN2B-G689C currents in *Xenopus* oocytes

With the intent to rescue (i.e. increase) current amplitudes, we turned our attention toward spermine— a naturally-occurring and highly specific GluN2B-subunit potentiator (*Mony et al., 2009*; *Mony et al., 2011*). We firstly assessed the effect of spermine (starting at its reported EC$_{50}$: 200 µM [*Mony et al., 2011*]) at physiological pH, specifically 7.3. Under these conditions, GluN2B*wt*-containing receptors underwent strong potentiation (~70%) by 200 µM spermine (hGluN2B-*wt* = 172% ± 7.8, n=14), and this potentiation gradually decreased the higher spermine was applied, consistent with the inhibitory effect of the reagent (*Figure 5a*, dark blue trace and b) (*Mony et al., 2011*; *Traynelis et al., 1995*). Surprisingly, spermine was much less effective over the variants under all concentrations tested, with G689S channels showing the least propensity to undergo potentiation (at 200 µM spermine- hGluN2B-G689C = 123% ± 4.0, n=16; G689S = 102.6% ± 3.5, n=15) (*Figure 5a*). In fact, spermine inhibited 2B-G689S channels at 500 µM, as well as 2B-G689C channels but at higher concentrations (1 mM). Notably, this effect was not observed for 2B*wt*-receptors (*Figure 5b*). To examine whether the weak potentiation, and further inhibition, of the variants by spermine resulted from changes in spermine's binding site (even though the binding domain is

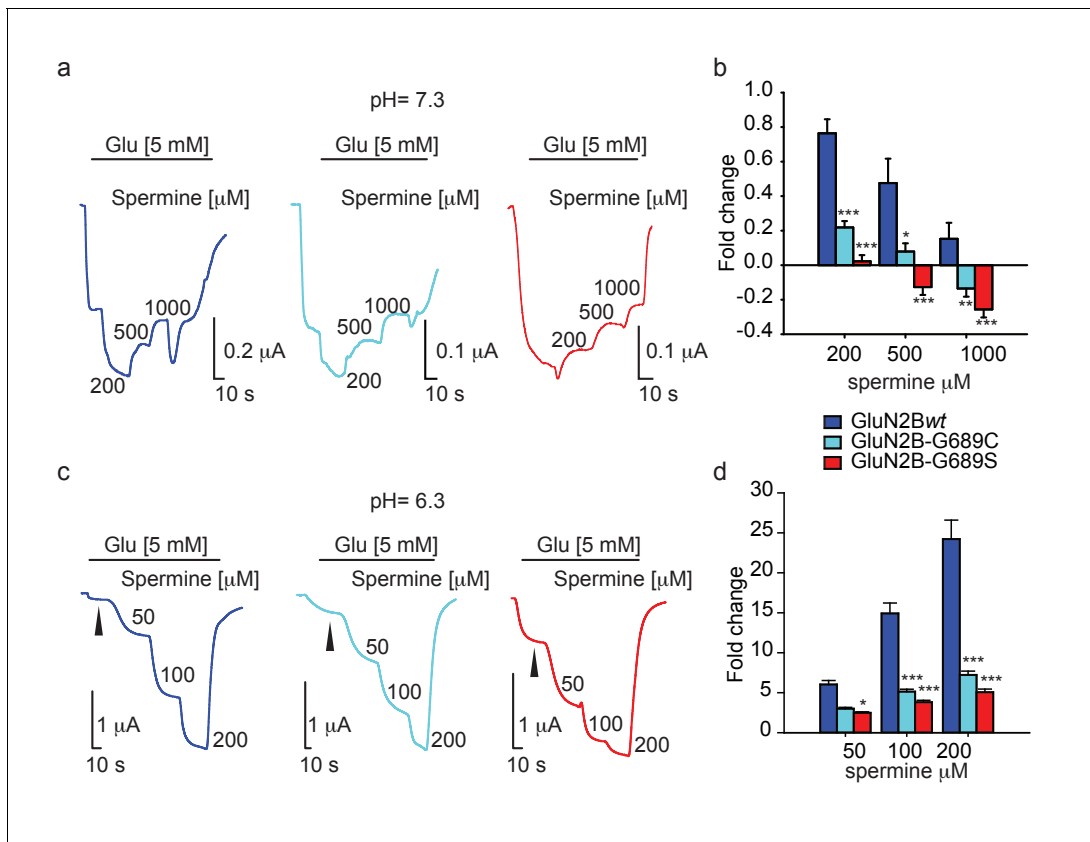

**Figure 5.** Spermine weakly potentiates variants *in Xenopus oocytes.* (**a and c**) Representative traces from oocytes co-expressing hGluN1a and hGluN2B-*wt* (blue) or GluN2B-G689C (cyan) or G689S (red), in response to increases in spermine concentrations (indicated next to trace in µM) after receptors activation (indicated by black line by 5 mM glutamate) at pH=7.3 (**a**) or pH 6.3 (**c**). At pH = 7.3, G689S failed to respond to spermine under all conditions and, instead, was inhibited by the drug. (**b and d**) Summary of spermine potentiation at pH=7.3 (**b**) or pH = 6.3 (**d**). Black arrowhead (**c**) shows the strongly diminished basal glutamate-current at pH=6.3; I$_{basal}$. In **c**, note that the variants (cyan and red) exhibit a larger I$_{basal}$ than oocytes expressing hGluN1a and hGluN2B-*wt* (blue), though it did not reach significance. This resulted from the application of 5 mM glutamate that saturates currents of hGluN1a and hGluN2B-*wt* (blue), but not of the variants. From two to three independent experiments, (**b**) n = 4–16 oocytes; (**d**) n = 14–20 cells. *, p<0.05; ***, p<0.001. Statistics show comparison to the corresponding GluN2B*wt* group (blue).

The online version of this article includes the following figure supplement(s) for figure 5:

**Figure supplement 1.** LBD mutations do not affect spermine binding site.

**Figure supplement 2.** Spermine potentiation is correlated with proton sensitivity of the variants.

thought to be located at the interface of the amino terminal domains of GluN1a and −2B subunits—very distant from he G689 residue [*Mony et al., 2011*]), we turned to assess spermine's effect at lower pH (pH 6.3), at which its effect is maximized (*Mony et al., 2011*; *Traynelis et al., 1995*). Indeed, GluN2B*wt*-containing receptors showed significantly larger potentiation by spermine, for instance ~25-fold at 200 μM of spermine (*Figure 5c*, dark blue trace, d; compare with *Figure 5b* 200 μM). Under these conditions, spermine did potentiate the currents of mutant channels in a dose-dependent manner (*Figure 5c,d*). These indicate that the binding domain of spermine remained intact. To further address this issue we employed arcaine-sulfate (a competitive antagonist of the polyamine site [*Gomes et al., 2014*; *Araneda et al., 1999*; *Reynolds, 1990*]). We applied arcaine at 200 μM (~3.5-fold above its IC$_{50}$[*Donevan et al., 1992*]) during the activation of the channels by saturating glutamate and glycine concentrations (Materials and methods). This treatment yielded equipotent inhibition (~90%) of all channel types (*Figure 5—figure supplement 1a,b*). Then, application of incrementing concentrations of spermine led to increases in current amplitudes of all channel types, showing spermine's expected capacity to displace arcaine (e.g., *Figure 5—figure supplement 1a*). However, potentiation remained largest for GluN2B*wt*-containing channels (*Figure 5—figure supplement 1c*). This observation supports our above results and demonstrates that the common binding site for arcaine-sulfate and spermine has not been altered by the mutations. However, it fails to explain the poor potentiation of the variants by spermine (*Figure 5d* and *Figure 5—figure supplement 1c*).

To address the latter, we re-examined our recordings and noted that, whereas all currents were significantly smaller at pH 6.5 (expectedly owing to GluN2B's pH-dependence [*Mony et al., 2009*; *Traynelis et al., 1995*; *Low et al., 2003*; *Jang et al., 2004*; *Banke et al., 2005*; *Williams, 2009*]), those of −2B*wt* receptors were consistently the smallest (*Figure 5c*, arrowheads and *Figure 5—figure supplement 1a*, arrowheads; summary in d). To ensure these differences did not stem from potential differences in expression (or even mRNA quality), we repeated the experiment with additional measurements of the total currents of each oocyte at pH = 7.3 (*Figure 5—figure supplement 2a*). We observed that −2B*wt* receptors exhibit both smallest and largest currents at pH = 6.3 and pH = 7.3, respectively (*Figure 5—figure supplement 2*, blue; summarized in b); better reflected by the pH-dependent current-ratio (I$_{7.3}$/I$_{6.3}$; *Figure 5—figure supplement 2c*, blue). The variants, on the other hand, displayed larger currents at low pH (*Figure 5—figure supplement 2a,b*, cyan, red), but smaller at pH 7.3 (G689C showing ~45% of maximal current as shown above; *Figure 2*), thereby smaller pH-dependent current ratios (*Figure 5—figure supplement 2c*). These results demonstrate that the activity of the variants is much less pH-dependent than GluN2B*wt*-containing receptors and provides an explanation why spermine fails to potentiate the channels (proton-inhibition is coupled to spermine potentiation for GluN2B-containing receptors [*Hansen et al., 2018*; *Mony et al., 2009*; *Mony et al., 2011*; *Traynelis et al., 1995*; *Low et al., 2003*; *Banke et al., 2005*]). Indeed, we find a positive correlation between pH-sensitivity (I$_{7.3}$/I$_{6.3}$) and spermine-potentiation for all channel types (*Figure 5—figure supplement 2d*). Together, this analysis shows that the variants do bind and respond to spermine (by a similar mechanism as 2B*wt*-containing receptors), but the lower magnitudes of potentiation suggest that the variants have reduced proton-sensitivity.

## GluN2B-G689C and G689S-receptors are resistant to proton inhibition

Proton-sensing in GluN2-subunits is thought to be contributed by a 'proton-senor' residing somewhere along the linkers connecting S2 and the transmembrane domains, though its exact location remains debated (and likely involves multiple regions in the subunit) (*Mony et al., 2009*; *Low et al., 2003*). Interestingly, several of the proposed locations are adjacent to the G689 residue. We conducted pH dose-response curves for all three channels and found that, as hypothesized (above), mutant channels exhibit a significant rightward shift (i.e. reduction) in pH-sensitivity (*Figure 6a,b*), and behind the reason why they fail to respond to spermine. These results also point toward the fact that the G689 residue is part of the GluN2B's proton-sensing domain. Although highly unlikely that the G689 residue directly binds protons (glycine is weakly titratable ), our results are in-line with reports showing the involvement of other small non-titratable amino acids (e.g. alanine and valine [*Low et al., 2003*]) in proton sensing (*Mony et al., 2009*). Thus, G689 is a new residue involved in proton sensing by the GluN2B subunit; accounting for the strongly reduced pH-sensitivity of the variants, larger currents at acidic pH and the reason for their reduced spermine sensitivity.

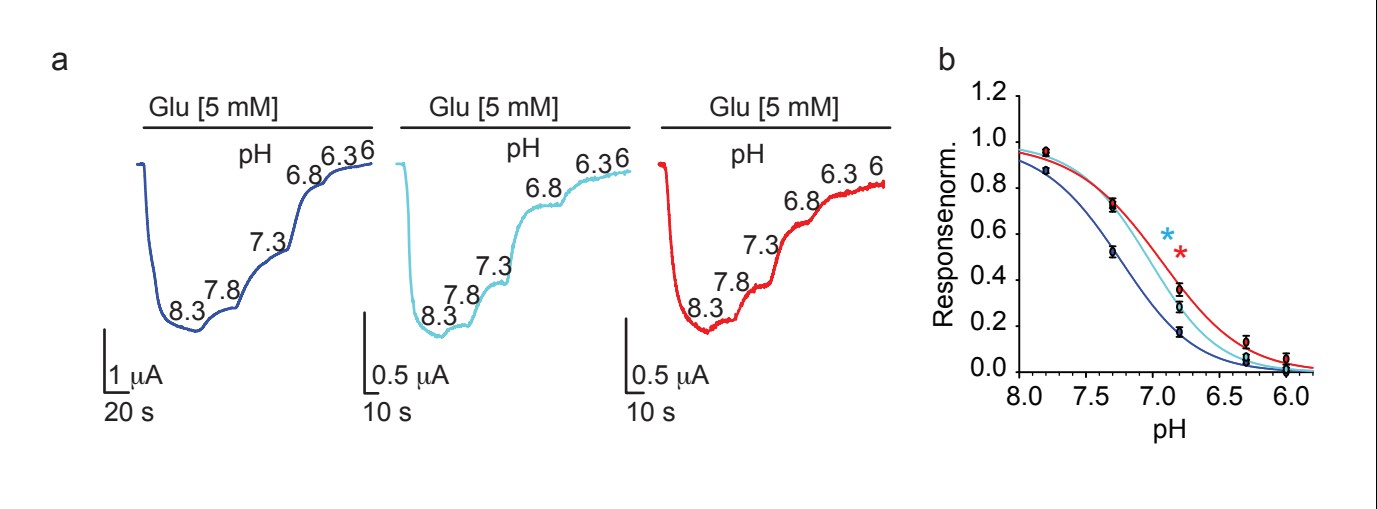

**Figure 6.** GluN2B variants show reduced proton-sensitivity. (a) Representative traces from oocytes co-expressing hGluN1a and hGluN2B-*wt* (blue) or GluN2B-G689C (cyan) or G689S (red), in response to decreases in pH (indicated next to trace in 8.3–6), summarized in (b). * (cyan- G689C and red, G689S) indicate $p < 0.05$ compared to $IC_{50}$ of *wt* (see **Table 2**). From N=two independent experiments, n = 15–21 cells.

## D-serine does not potentiate GluN2B-currents

With the failure of spermine to potentiate the mutant channels, we proceeded to test another suggested channel potentiator, D-serine (**Soto et al., 2019**). Briefly, Soto et al. showed the augmentation of glutamate currents (of GluN2B*wt*-containing receptors and a GluN2B-mutant with a ~7-fold reduction in $EC_{50}$) by direct application of D-serine, or by *in vivo* supplementation of L-serine (which converts to D-serine [**Neame et al., 2019**]). To test whether this could be a potential treatment in our cases, we first tested the potentiation nature of D-serine. We applied 5 mM glutamate (to maximally open the channels) along three different glycine concentrations (over three orders of magnitude). To each ligand combination, we added a constant concentration of D-serine (100 μM; shown to exert the greatest potentiating effect **Soto et al., 2019**; **Figure 7**). At low glycine concentrations (1 μM), the addition of D-serine increased the maximal current of the 2B*wt* receptors by ~35%, as well as the currents of the variants, and even to a larger extent (~60%)(**Figure 7a,b**). However, D-serine poorly augmented the currents when glycine was added at 10 μM and showed no augmentation in the presence of 100 μM (saturating) glycine (**Figure 7b**). These observations strongly suggests that D-serine is not a *bona fide* potentiator, rather increase in current amplitudes results from saturating the GluN1a subunit (see glycine dose-responses, **Figure 2—figure supplement 2a,b** and Discussion). Next, we assessed the effect of D-serine, albeit at physiologically relevant sub-saturating glutamate and, more importantly, glycine concentrations (**Zhang et al., 2018**). In the case of G689C, application of D-serine (on top of 1 μM glycine) yielded a ~2-fold increase in the glutamate current, however this current represents <15% of the total current that can be obtained by fully opened receptors (i.e. at 5 mM glutamate) (**Figure 7c,d**; cyan-filled bars). Notably, G689S channels did not respond to physiological concentrations of glutamate/glycine and thus did not show any responses to D-serine (**Figure 7c, d**; red-filled bars). Thus, the effect of D-serine over these severe LoF mutations is negligible, and completely absent in the case of G689S. We therefore do not recommend the use of L-serine as a treatment for these, and potentially other variants, exhibiting extreme LoF, as the benefits (i.e. subtle increases in current of the variants) may not exceed the potential side-effects that may ensue by the non-specific activation of *all* other GluNRs in the brain by D-serine.

## Major reduction in glutamate potency reconstituted in mammalian cells and dominant-negative effect in neurons

We next turned to assess glutamate potency in mammalian cells, specifically HEK293T cells. As in oocytes, we co-expressed hGluN1a with the different GluN2B-subtypes and performed dose-response curves using patch clamp (Materials methods). We obtained similar $EC_{50}$ values for GluN2B*wt* and 2B-G689S-containing receptors, on par with values obtained in oocytes (~2 mM;

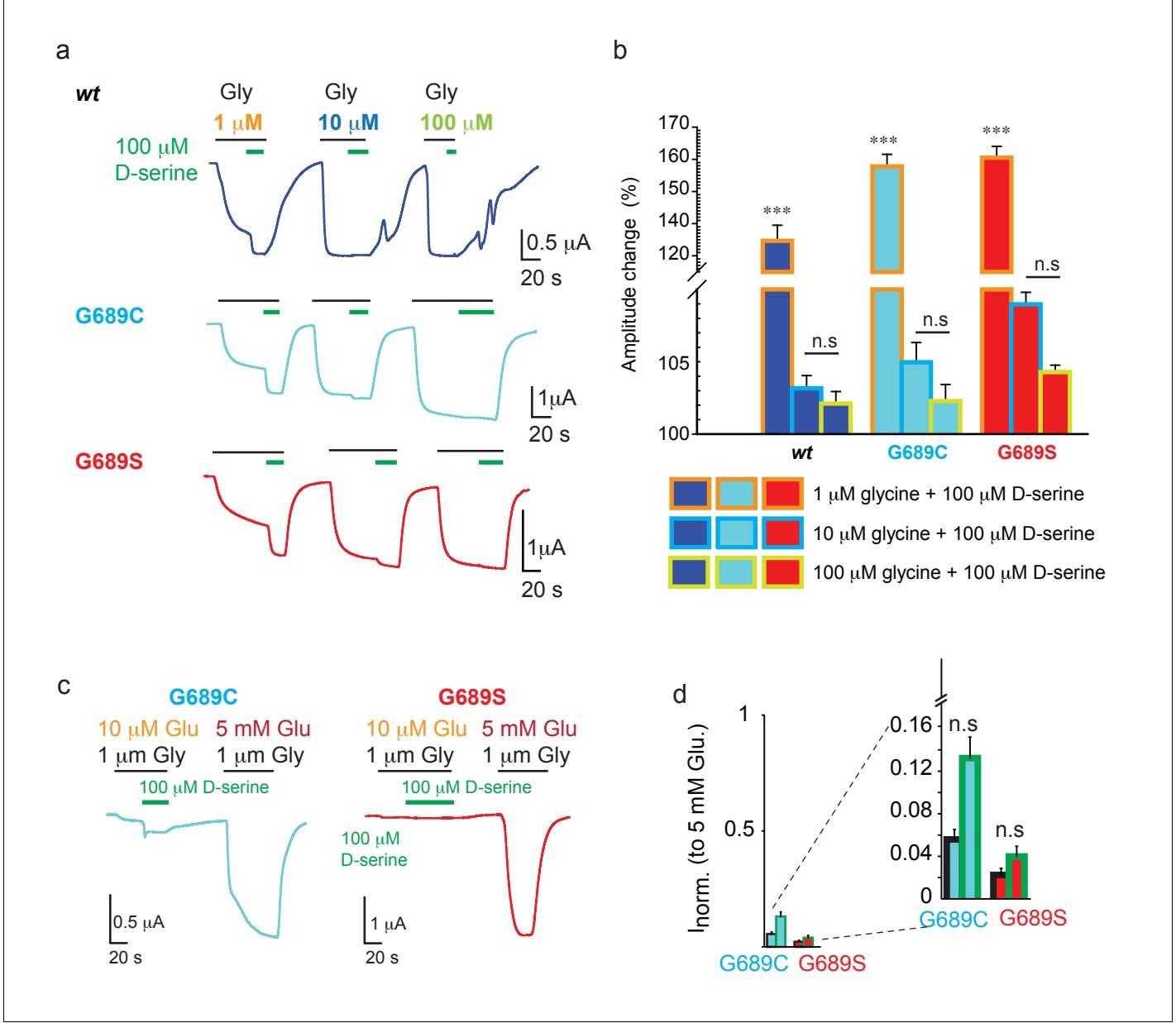

**Figure 7.** D-serine does not potentiate GluNRs. (a) Representative traces from oocytes co-expressing hGluN1a and hGluN2B-*wt* (blue) or GluN2B-G689C (cyan) or G689S (red) in response to 100 μM D-serine (green bar above traces) after receptors have been activated by 5 mM glutamate along 1 (orange), 10 (light blue), or 100 μM (light green) glycine. **b** Summary of changes in amplitude after application of 100 μM (color coded as in a). **c** Representative traces and assessment of the effect of application of 100 μM D-serine (as shown in a), but when channels are activated by physiologically relevant glutamate (10 μM) and glycine (1 μM) concentrations. Total current of channels (and proof of expression) was obtained by application of 5 mM glutamate before end of recording. D-serine further activates, albeit weakly, G689C (cyan), but has no effect over 2B-G689S receptors (red); summarized in (d) and inset (zoom in on region of plot). In (d), black outline of bars represents currents obtained by 1 μM glycine. Green outline represents the currents obtained by 1 μM glycine and D-serine. *** indicates p< 0.001 compared to other glycine concentrations for the same variant. n.s., not significant. hGluN2B-*wt*, n=10; hGluN2B-G689C, n=11; hGluN2B-G689S, n=12. For (d), n = 8 (G689C), n=7 (G689S).

*Figure 8—figure supplement 1* and *Table 2*). GluN2B-G689C failed to express at sufficient levels for precise current measurements. Here, too, we did not exceed 10 mM as application of higher concentrations of glutamate tended to yield non-specific responses (*Figure 8—figure supplement 1a*, black trace). Thus, these observations are consistent with our expression assays (see *Figure 3*) and TEVC measurements (*Figure 2*, *Table 2*). Additionally, these results also highlight the advantage in

using oocytes for biophysical characterization of *GRIN* mutations, in particular mutations that reduce membrane expression levels.

We next overexpressed the variants (without over-expression of GluN1) in cultured rat primary hippocampal neurons, in which we examined synaptic activity by patch clamp (*Figure 8a*). Each neuron was initially recorded under conditions isolating α-amino-3-hydroxy-5-methyl-4-isoxazolepropionic acid receptors (AMPARs; GluARs)-dependent miniature EPSCs (minis_AMPAR), followed by recording of GluNR (NMDAR)-dependent minis (minis_NMDAR)(see Materials and methods). Neurons overexpressing the variants did not show any differences in morphology, membrane resistance, capacitance, and resting potential (*Figure 8—figure supplement 2a,b*). However, overexpression caused a strong reduction in synaptic GluNR-events (*Figure 8b*). More specifically, neurons overexpressing the variants showed a strong reduction in the frequency of mini_NMDAR, but with unaffected frequencies of mini_AMPAR (*Figure 8b*, asterisks); yielding a ~50% reduction in the mini_NMDAR/AMPAR ratio compared to control group (*Figure 8c,d*). Interestingly, solely the overexpression of G689S induced a significant reduction in the amplitude of mini_NMDAR along an increase in the amplitude of mini_AMPAR (*Figure 8—figure supplement 2c*). Overexpression of G689C caused a small, albeit significant, increase in mini_NMDAR's amplitude (*Figure 8—figure supplement 2c*, right panel). Importantly, mini_NMDARs from neurons overexpressing the variants displayed faster deactivation kinetics than control (*Figure 8d*). These demonstrate that overexpression of the variants in hippocampal neurons prompts a pronounced effect on synaptic GluNRs. The reduction in the frequency of

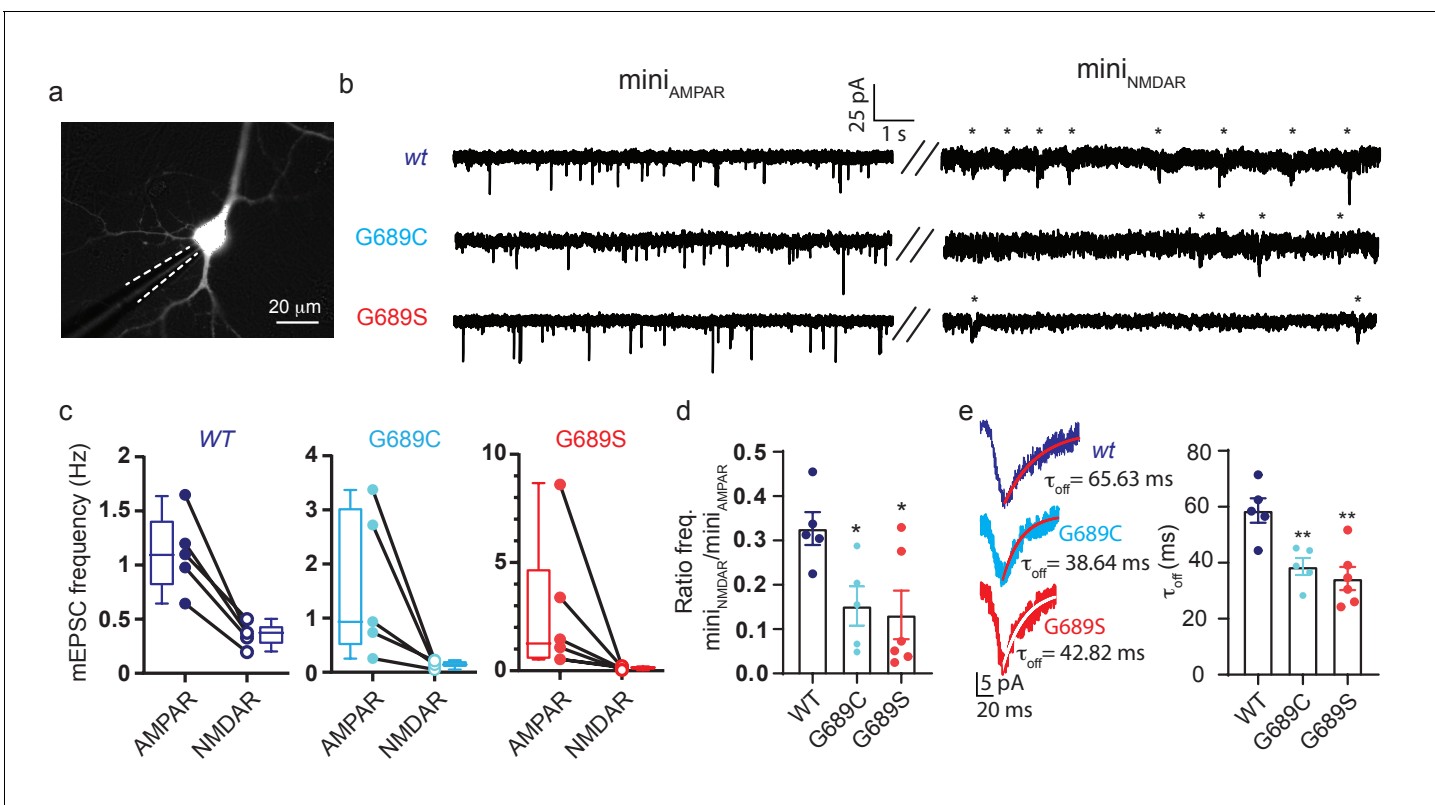

**Figure 8.** GluN2B variants modulate synaptic glutamatergic activity. (**a**) Representative micrograph showing a transfected neurons (expressing GluN2B and YFP) and a recording pipette (dashed white line). (**b**) Representative recordings of miniature EPSCs from neurons overexpressing GluN2B*wt* (top), GluN2B-G689C (middle) or GluN2B-G689S (bottom) subunits. Left- recordings of mini_AMPAR; right- mini_NMDAR (highlighted by asterisks). (**c**) Frequencies of mini_AMPAR and mini_NMDAR for each cell, summarized in (**d**). (**d**) Ratio of NMDAR to AMPAR minis' frequency. (**e**) mini_NMDAR show reduced decay kinetics. Traces are averaged miniature EPSCs from events as shown in (**b**), fitted with a mono-exponential fit (red), summarized in plot (τ_off). n.s., not significant. *, p<0.05 compared to control group, 2B*wt*. hGluN2B-*wt*, n=5; hGluN2B-G689C, n=5; hGluN2B-G689S, n=6. Fit (red)- exponential.

The online version of this article includes the following figure supplement(s) for figure 8:

**Figure supplement 1.** GluN2B-G689S shows reduced glutamate potency in mammalian cells.

**Figure supplement 2.** GluN2B variants modulate miniature EPSCs' amplitudes.

mini$_{NMDAR}$—in combination with the unaffected frequency of mini$_{AMPAR}$— rules-out loss of excitatory synapses (as may be instigated by other variants, for example GluN2B-S1413L [*Liu et al., 2017*]). Moreover, the mirroring changes in the amplitudes of both mini-types, following G689S-overexpression, is highly reminiscent of synaptic scaling induced by the strong decrease in responsive GluNRs at the synapse (*Figure 8—figure supplement 2c*), during which persistent block of GluNRs causes increases in synaptic GluARs (*Sutton et al., 2006*). Lastly, the reductions in τ$_{off}$ due to overexpression of the variants strongly suggests that the remaining current is contributed by the faster deactivating GluN2A-subunits (*Figure 8d*; *Paoletti et al., 2013*). Notably, these effects are consistent with our dominant-negative observations (see *Figure 4*) and results obtained from animal model bearing a GluN2B LoF mutation (*Shin et al., 2020*).

## Discussion

Here, we describe, and characterize, two *de novo* heterozygous *GRIN2B* loss-of-function (LoF) variants—G689C and G689S—in two pediatric patients (*Figure 1a*). Both patients display severe DD, ID, dyskinetic movements, and speech impairment (*Table 1*); four very well-documented phenotypes observed in *GRIN2B* patients (*Myers et al., 2019*; *Platzer et al., 2017*). We also find that the G689S-patient exhibits epileptic seizures, consistent with the two previously documented G689S-patients (*Platzer et al., 2017*). Interestingly, the G689C-patient does not show seizures, but the clinical likelihood of this young patient to develop epileptic seizures remains relatively high (*Myers et al., 2019*; *Platzer et al., 2017*; *Devinsky et al., 2018*). Along with the known comorbidities of *GRIN2B* patients, here we report several phenotypes that deviate from reported symptoms. The G689S patient displays hypertonia, rather than hypotonia (*Myers et al., 2019*; *Platzer et al., 2017*), as is also displayed by the G689C patient. Moreover, while movement disorders are common in *GRIN* patients, the G689C patient (but not the G689S patient) is also reported to exhibit hypermobility/hyperflexibility. The reason for this divergence between these two synonymous mutations is poorly understood (as both mutations are highly analogous and instigate similar LoF). However, it is highly common for *GRIN*-patients with similar (or even identical) mutations to show diversifying symptoms (*Myers et al., 2019*; *García-Recio et al., 2021*; *Platzer et al., 2017*; *Strehlow et al., 2019*; *Supplementary file 2*) or, inversely, for patients with different mutations (even in different *GRINs*) to display overlapping symptoms (*Myers et al., 2019*). These phenotypes are likely to involve additional genetic and environmental factors. In the case of the G689S patient, we do find (by whole-exome sequencing; WES) two additional variants of unknown/uncertain significance (VUS), specifically the *SLC6A8* (creatine Transporter) and *CACNA1A* (voltage-gated calcium Channel) genes. However, whether these are directly involved or implicated in the disease remains unknown. Nonetheless, no VUS, pathogenic variants, nor aberrations in the Chromosomal Microarray Analysis (CMA) were detected in screens of the G689C patient (*Table 1*). Resultantly, we and others suggest that patients' symptoms cannot serve as primary diagnostic measures for *GRIN*-related encephalopathies (e.g. [*Zehavi et al., 2017*]). Indeed, no formal diagnostic criteria for *GRIN2B*-related neurodevelopmental disorder have been established (*Platzer and Lemke, 1993*; revised 2021).

An additional puzzling observation shows that the severity of phenotypes does not correlate with the magnitude of the effect of the mutation on channel function (*Tang et al., 2020*). In fact, despite the critical LoF effect of the mutations described here, the clinical phenotypes resemble many other *GRIN2B* LoF mutations (*Supplementary file 1*; *Swanger et al., 2016*; *Bell et al., 2018*). For instance, GluN2B-E413G inflicts a much smaller reduction in glutamate potency (EC$_{50}$ = 79 ± 5.3 μM), though manifests with severe phenotypes as described here for the two variants. This dissonance is reflected in a recent *GRIN2B* study (consisting of a cohort of 86 patients), in which they found no clear association between: (1) effect of mutation (GoF or LoF), (2) extent of effect (e.g. shift in EC$_{50}$), and (3) localization of the mutations in the subunit, with the clinical outcomes (*Platzer et al., 2017*). The only significant correlation obtained was between variant class (i.e. missense or truncation mutation) and intellectual outcome (mild to moderate moderate vs severe ID); with truncation carriers *tending* to present mild/moderate intellectual disability. However, it is not really unexpected for these symptoms to appear in *GRIN2B* cases, owing to the subunit's very early (embryonic) expression pattern (*Paoletti et al., 2013*; *Platzer et al., 2017*; *Myers et al., 2019*). In contrast, analysis of a larger cohort of *GRIN2A* patients (n = 248) finds only *two* distinct phenotypes that could be associated with the location and effect (gain or loss—but not size of effect) of the

mutations (*Strehlow et al., 2019*). More precisely, the authors show that: (1) mutations in TMDs and connecting linkers yield GoF (although to very different extents) and these may be associated with broad developmental and epileptic encephalopathy phenotypes. (2) Mutations in the two large extracellular domains (ATD, LBD) typically instigate LoF and are better related with speech abnormalities and seizures, with mild to no ID (*Strehlow et al., 2019*). These observations highlight the need for larger cohorts in order to establish better correlations (if any).

Despite these associations, it remains challenging to infer how, and to what extent, a mutation may affect receptor function or its expression solely based on clinical symptoms or even based on the location of the mutation within the protein (*XiangWei et al., 2018*). In our case, *in silico* scrutiny of the G689 residue shows it to reside at the lower lobe of the LBD of the GluN2B subunit (S2 domain)—at the base of the glutamate entry tunnel—virtually lining the glutamate binding pocket (*Figure 1b* and *Figure 1—figure supplement 1*). When the location of the residue is combined with structure simulations of the variants (*Figure 1c,d*, *Figure 1—figure supplement 1b* and *Figure 1—figure supplement 2*), estimation of protein stability (ΔΔG; *Figure 1—figure supplement 4*), and considering that most GluN2B mutations in LBD reduce glutamate potency (*Supplementary file 1* and *2*; *XiangWei et al., 2018*; *Tang et al., 2020*; *Platzer et al., 2017*; *Swanger et al., 2016*; *Vyklicky et al., 2018*), it was safe to assume that the variants would yield LoF. However, these could not have predicted the extent of the effect, explicitly >1000-fold and ~2000-fold by G689C and G689S, respectively (*Figure 2*, *Figure 8—figure supplement 1* and *Table 2*). Potential reasons why we could not anticipate these large shifts are likely because they represent two very extreme cases (only second to another potent *GRIN2A* mutation D731N; ~6000-fold reduction [*Swanger et al., 2016*]) and because of the very few characterized LBD-mutations in *GRIN2B* (~20 mutations, See *Supplementary file 1* and *2*). With these limitations in mind, we decided to explore the functional outcome of any possible mutation within the LBD (consisting of ~280 a.a.); yielding a collection of 5282 substitutions. This large number made it impractical for us to employ structure simulations, thereby motivating us to proceed with ΔΔG estimations (*Pires et al., 2014*). We calculated ΔΔG for 5282 substitutions within the LBD of GluN2B (*Supplementary file 3*, closed circles) and immediately note that most of the substitutions yield a negative outcome on protein stability (ΔΔG < −0.5; *Supplementary file 3a*, red region). Substitutions with a stabilizing nature appeared in a much smaller fraction of cases (~20%) (*Supplementary file 3a, b* -blue). We explored the relationship between characterized LBD mutations with our ΔΔG estimates, however there are too few characterized mutations (18 mutations) to address this (*Supplementary file 3c*). However, we can observe that LoF mutations correlate better with more negative ΔΔG, whereas GoF mutations are associated with less negative ΔΔG (albeit this is based on two out of three characterized mutations) (*Supplementary file 3c*). To examine whether there are locations within the LBD that might be more vulnerable for mutagenesis, we estimated each residue's relative contribution to the stability of the LBD (by averaging the ΔΔG of 19 substitutions for each residue, see Materials and methods) and plotted these on the structure. Unfortunately, we do not observe negative hot-spots nor did the 18 characterized mutations show a pattern in the LBD (*Supplementary file 3d*). However, the data does suggest that stabilizing mutations are likely to locate on the outer layers of the LBD (*Supplementary file 3e*, blue and dashed blue circle). Indeed, these estimations encompassed all three GoF mutations. Interestingly, and perhaps with the most predictive nature, this analysis emphasizes that mutations resulting in a glycine or a serine—no matter the residue they replace in the LBD—are likely to be the most damaging (*Supplementary file 3f*). This is highly reasonable as these amino acids are very small (*Figure 1—figure supplement 4*), especially glycine, and their incorporation *in-lieu* of other (larger) residues should be very destabilizing. Interestingly, the same may apply inversely, namely removal of glycine and its substitution by other (larger) residue should be highly disfavorable for receptor function. In support, glycine residues are suggested to serve as essential hinges in *GRIN2B* and their mutagenesis causes severe channel dysfunction (*Amin et al., 2018*), including the two cases described here.

Another elusive feature for prediction is expression of variants, especially when the mutations do not lead to truncation (*García-Recio et al., 2021*). It has been previously shown that high glutamate affinity is associated with proper surface trafficking (*She et al., 2012*; *Horak et al., 2014*; *Wang et al., 2015*; *Kenny et al., 2009*). Thus, it can be assumed that since most LBD mutations yield reduction in EC$_{50}$, most should also show reduced expression levels. This assumption correctly predicts the effect of the G689C mutation (*Figure 3a,b*), but completely fails to explain the robust

expression of the G689S variant with an even larger reduction in $EC_{50}$ (*Figures 2* and *3*, *Figure 8—figure supplements 1* and *2*). This assumption (*She et al., 2012*; *Vyklicky et al., 2018*) further neglects to explain how both low-affinity variants enhance the expression of the *wt* subunit (*Figures 3c* and *4* and *Figure 4—figure supplement 1*). Thus, our results describe two novel cases in which intracellular glutamate-binding is not essential for proper trafficking of GluN2B-subunits to the membrane and cautions the use of glutamate affinity for predicting expression levels and/or functional effect. Together, the functional outcome of our observations suggests that during normal synaptic transmission, tri-heteromeric receptors assembled from GluN2B*wt* and mutant variants, are non-responsive to normal neurotransmission (*Figures 4* and *8*; *Lisman et al., 2007*). In fact, regular neurotransmission does not typically saturate *wildtype* receptors (*Ishikawa et al., 2002*; *Nimchinsky et al., 2004*). Thus, we suggest that the severe clinical phenotypes observed for both variants arise from a complex combination of: (1) severe LoF of channel consisting of two copies of the variants, (2) poor trafficking to membrane, (3) co-assembly and exertion of dominant-negative effect over native GluN2B*wt*-containing receptors.

With the intention to provide a potential treatment, we examined whether spermine would potentiate the currents of the variants. Of note, we focused on spermine as it is highly GluN2B-selective (*Mony et al., 2011*), unlike other potentiators (e.g. tobramycin, pregnenolone-sulfate; PS or D-serine) that can exert non-specific effects on other GluN-subunits or even other glutamate-receptors (*Tang et al., 2020*; *Swanger et al., 2016*; *Stoll et al., 2007*; *Masuko et al., 1999*). Secondly, spermine can cross the blood-brain-barrier to certain extents (BBB [*Shin et al., 1985*; *Diler et al., 2002*]), and could potentially be administered orally or intraperitoneally (e.g. *Okumura et al., 2016*; *Guerra et al., 2016*). Third, it is inexpensive. Importantly, previous reports showed that spermine acted on LoF GluN2B variants in a similar fashion as they do on GluN2B*wt*-subunits (*Swanger et al., 2016*). However, and strikingly, we found that G689C- and G689S-containing channels poorly respond to the reagent under physiological conditions; with the G689S-mutant even undergoing strong inhibition by spermine (*Figure 5a,b*). We go on to demonstrate that reduced spermine-sensitivity stems from the variants' reduced pH-sensitivity (*Figure 6* and *Figure 5—figure supplement 2*) (but not disrupted binding domain; *Figure 5—figure supplement 1*). This suggests that the G689 residue is directly involved in proton-sensing in the GluN2B subunit (*Low et al., 2003*; *Banke et al., 2005*; *DeCoursey, 2018*). This feature adds another layer of complexity to the growing list of effects exerted by these unique mutations. These observations suggest that though spermine (or potentially other GluN2B-positive allosteric modulators [*Mony et al., 2009*; *Burnell et al., 2019*; *Zhu and Paoletti, 2015*]) may be useful in other cases of GluN2B-LoF mutations, it is not suitable for treating G689C- or G698S-induced deficiencies at the synapse.

We similarly tested another suggested potentiator, namely L- (but *de facto* D)-serine (*Soto et al., 2019*). D-serine was suggested to act as potentiator owing to its ability to increase currents of a mild LoF *GRIN2B* variant (showing ~seven-fold reduction in $EC_{50}$). We applied 100 μM D-serine onto receptors activated by sub- or saturating glycine concentrations (*Figure 7*), but D-serine enhanced the currents of the variants solely when glycine concentrations were sub-saturating (see glycine $EC_{50}$, *Figure 2—figure supplement 2*). Thus, it appears that D-serine does not act as a classic potentiator (i.e. [*Tang et al., 2020*; *Mony et al., 2009*; *Burnell et al., 2019*; *Hackos et al., 2016*]). Instead, it is an equipotent ligand for the GluN1a subunit and the observed increases in currents (up to 60%) are obtained by saturation of GluN1 (glycine or D-serine; $EC_{50}$: ~0.7 [*Traynelis et al., 2010*]). Regardless the exact definition of the mechanism by which D-serine augments the currents, under physiological conditions increase in the extracellular D-serine concentration would likely lead to further opening of the channels, as resting glycine (and D-serine) extracellular concentrations may be sub-saturating; 1–5 μM (*Zhang et al., 2018*; *Hashimoto et al., 1995*; *Lazarewicz et al., 1992*). However, and even in cases where GluN1a subunits are not fully saturated by glycine, the G689C or G689S mutations require very high (mM) glutamate concentrations for opening and, therefore, increase in D-serine (via L-serine supplementation) is ineffective (*Figure 7c,d*). We therefore do not recommend the use of L-serine in this, or other extreme LoF mutations, as L-serine may acts on the obligatory GluN1-subunit found in *all* receptor combinations and possibly induce side effects.

In summary, we have systematically characterized two unique mutations occurring at the same residue of the *GRIN2B* gene in two patients. The variants exert a strong dominant-negative effect over *wt*-subunits, leading to reduced potency of mixed channels and reduced synaptic GluNR-currents (and compensation by AMPARs [*Sutton et al., 2006*]). To make things worse, the variants are

resistant to protons, thereby limiting the use of spermine. We provide an assessment of the stability of the LBD for all possible mutations within the LBD. These exemplify the vulnerability of the LBD to mutagenesis, particularly for insertion (but also deletions) of glycine and serine. Together, our study complements ongoing efforts invested in characterizing *GRIN* variants (appearing faster than can be functionally tested, *Supplementary file 2*), provide insights concerning the structure-function relationship of GluN2B and underscore the need for new, more potent, GluN2B-specific channel openers.

## Materials and methods

### *Xenopus* oocytes extraction

All Experiments were approved by the Technion Institutional Animal Care and Use Committee (permit SB, no. IL-129-09-17). *Xenopus laevis* oocytes were harvested, prepared, and injected with mRNA as previously described (*Berlin et al., 2020*). Briefly, oocytes were obtained from anesthetized (by 0.4% tricaine solution) female frogs. Oocytes were extracted from ovaries, defolliculated by collagenase treatment in ND96 $Ca^{+2}$-free solution (in mM: 96 NaCl, 2 KCL, 1 MgCl2, 5 HEPES, pH 7.4) for about 45 min at RT. Cells were washed with ND96 $Ca^{+2}$ -free and transferred to enriched ND96 (NDE) consisting of ND96 supplemented with 1.8 mM $CaCl_2$, 2.5 mM sodium pyruvate, 100 µg/ml streptomycin and 62.75 µg/ml penicillin. Stage V oocytes were manually isolated, incubated overnight at 18°C and injected with mRNA the next day.

### Mammalian cell culture and transfection

HEK293T cells were maintained in DMEM (containing 10% FBS and 1% L-glutamine) in 100 mm Corning cell culture dishes. Cells were purchased from the American Tissue Culture Collection (ATCC) and are regularly tested for mycoplasma. A day prior transfection, medium was replaced by 2 mL of fresh medium and cell were suspended (by gentle pipetting). Suspended cells were then transferred (~250 µl) onto a 35 mm cell culture dish containing 2 ml medium. Cells were grown to 70–90% confluency (~overnight). The following day, cells were transfected with ViaFect Transfection Reagent (Promega) by DNA mixtures: 400 µl serum free DMEM+ 1 µg DNA of hGluN1a-*wt* + 1 µg hGluN2B variant + 2 µg EGFP; at 3:1 ratio of transfection (µl reagent: µg DNA, here 12 µl reagent/4 µg total DNA). Reaction mix was allowed to settle at room temperature for 20 min, prior addition to cells. Following 4–8 hr, old medium (containing ViaFect reaction mix) was replaced by 2 mL of fresh DMEM-containing 10% FBS and 1% L-glutamine. Then, cells were resuspended and transferred into a new 24 well plate, containing poly-D-lysine (PDL)-coated coverslips (12 mm), and grown overnight with broad GluNR blockers, explicitly 2 µM MK-801 (Alomone labs, Cat. #M-230) and 200 µM AP5 (Alomone labs, Cat. #D-145).

### Dissociation, culturing, maintenance, and transfection of primary neurons

Primary cultures of hippocampal neurons were done as previously reported (*Berlin and Isacoff, 2018*). Briefly, hippocampi were extracted from rat neonates (P0), dissociated and transferred to 24-well plates containing PDL-covered glass coverslips containing MEM (Gibco)-based growth medium and kept in an incubator (37°C and 5% $CO_2$). After 5 days in vitro (DIV), the neuronal culture medium was supplemented with 4 µM cytosine arabinoside (ARA-C) for suppression of glial cell proliferation. At 7–9 days in vitro (DIV), neurons were transfected by the calcium-phosphate method with 0.3 µg DNA eYFP and 1 µg DNA of GluN1a and GluN2B*wt* or 2B-G689C or 2B-G689S. Neurons were then grown for 3–4 more days prior to recordings.

### Molecular biology and in vitro mRNA preparation

Human GluN1a-*wt* (hGluN1a), human GluN2B-*wt* (hGluN2B*wt*) and human GluN2B-G689C (hGluN2B-G689C) cloned in pCl-Neo were obtained from Dr. Garin-Shkolnik T (produced by the Traynelis S. Lab). Human GluN2B-G689S (hGluN2B-G689S; c.G2065A) was generated by us using site-directed mutagenesis (primers: sense- 5'-CCGCTTTGGGACCGTGCCCAACAGCAGCACAGA-GAGAAATATTCG-3', antisense 5'-CGAATATTTCTCTCTGTGCTGCTGTTGGGCACGG TCCCAAAGCGG-3') and verified by full DNA sequencing (Faculty of Medicine, Biomedical Core

Facility- Technion). For mRNA preparation, DNA plasmids were linearized by restriction enzymes (NotI), followed by in vitro mRNA transcription using mMessage-mMachine T7 kit (Thermo Scientific, Cat.#AM1344), as previously described (*Berlin et al., 2010*). mRNA concentrations were determined using a spectrophotometer. Oocytes were injected with 5–16 ng/oocyte mRNA of each subunit at 1:1 ratio in all of the experiments. For dominant-negative experiments, we co-injected hGluN1a, hGluN2B$wt$ and hGluN2B-G689C or −2B-G689S with the following mRNA quantities (ng/oocyte): 5:16.6:1, 5:5:5: and 5:1:16.6 yielding GluN2B-$wt$ high ($wt_H$), even ($wt_E$), and low ($wt_L$), respectively.

## Electrophysiology (TEVC and Patch clamp recordings)

Two electrode voltage clamp (TEVC) recordings were carried-out 24–72 hr following mRNA injections, as previously described (*Berlin et al., 2011*). Recordings were performed using commercial amplifier (Warner Instruments, USA) and Digitizer (Digidata-1550B; Molecular Devices, USA), controlled by pClamp10 software (Molecular Devices, USA). Electrodes consisted of pulled glass capillaries (by puller- Narishige, Japan) with chlorinated silver wire, filled with 3M KCl. Oocytes were clamped to −60 mV and perfused with nominally $Mg^{+2}$-free barth solution (in mM): 100 NaCl, 0.3 $BaCl_2$, 5 HEPES, pH 7.3 (adjusted with NaOH). For glutamate dose-response curves for the hGluN2B$wt$-containing channels glutamate concentrations ranged between 0.2 and 100 µM, including glycine at saturating concentration (100 µM). For the hGluN2B-G689C or G689S variants, glutamate concentrations arranged between 0.2 µM and 100 mM, in the presence of saturating glycine (100 µM). The different barth solutions containing excessively large glutamate concentrations were adjusted for osmolarity. Barth solution with glutamate concentrations ranging between 0.1 and 2 mM contained (in mM): 100 NaCl, 0.3 $BaCl_2$, 5 HEPES, 99 NMDG, pH 7.3 (adjusted with HCl). For higher concentrations of glutamate (>5 mM), recording solution contained (in mM): 100 NaCl, 0.3 $BaCl_2$, 5 HEPES, 90 NMDG, pH 7.3 (adjusted with HCl) or 100 NaCl, 0.3 $BaCl_2$, 5 HEPES, 50 NMDG, pH 7.3 (adjusted with HCl) were used in making 10- and 50 mM glutamate solutions, respectively. For barth containing 100 mM glutamate, we used NMDG-free solution (in mM): 100 NaCl, 0.3 $BaCl_2$, 5 HEPES, pH 7.3 (adjusted with NaOH). For glycine dose-response experiments, glycine solutions ranged between 0.05 and 100 µM supplemented with 5 mM glutamate were used. For $Mg^{+2}$-dose-response experiments, receptors were activated by 5 mM glutamate and 100 µM glycine, then gradually blocked by incrementing $Mg^{+2}$-concentrations ranging from 1 µM to 10 mM.

## HEK293T

Patch clamp recordings were acquired MultiClamp 700B and Digidata 1440A (Molecular Devices), as previously described (*Berlin and Isacoff, 2018*). Briefly, cells were voltage-clamped at −70 mV. Borosilicate glass capillaries (i.e. pipettes) were pulled to resistances of 4–10 MΩ and were filled with an internal solution containing (in mM): 135 K-gluconate, 10 NaCl, 10 HEPES, 2 $MgCl_2$, 2 $Mg^{2+}$-ATP, 1 EGTA, pH = 7.3. Recordings were done in extracellular recording solution containing (in mM): 138 NaCl, 1.5 KCl, 2.5 $CaCl_2$, 10 D-glucose, 5 HEPES, 0.05 glycine, pH = 7.4.

## Neurons

YFP-positive 10–13 DIV neurons were visually detected by 488 nm LED illumination (X-Cite fluorescence LED illuminator, Excelitas Technologies) and voltage-clamped at −80 mV (Multiclamp 700B amplifier and Digidata 1440A digitizer). Glass capillaries were adjusted to resistances of 8–12 MΩ, and filled with intracellular solution containing (in mM): 135 K-gluconate, 10 NaCl, 10 HEPES, 2 $MgCl_2$, 2 Mg-ATP, 1 EGTA, pH = 7.3. For recordings of AMPAR-mediated mEPSCs (mini$_{AMPAR}$), neurons were perfused with an extracellular solution containing (in mM): 138 NaCl, 1.5 KCl, 1.2 MgCl2, 5 $CaCl_2$, 10 Glucose, 5 HEPES, pH = 7.4, and 1 µM TTX. For recordings of NMDAR-mediated mEPSCs (mini$_{NMDAR}$), cells were perfused with the extracellular solution enriched with 5 µM Glycine, 1 µM TTX and 20 µM CNQX (selective AMPAR blocker) but without $MgCl_2$. Cells were recorded for ~3 min for each perfusion phase.

## Apparent open probability

MK-801 (activity-dependent pore blocker of GluNRs) was purchased from Alomone labs (cat.# M-230). One mM stock solutions were made by diluting lyophilized MK-801 in barth solution. Receptors were activated by 3 mM glutamate and 100 µM glycine solution, then blocked by 1 µM MK-801

solution containing both agonists. Deactivation kinetics were fitted from which we extracted $t_{10-90\%}$ or deactivation constant, $\tau_{off}$.

## Spermine potentiation

Spermine was purchased from Sigma-Aldrich (Cat. #S3256). A total of 200 mM stock solutions were made by diluting powdered spermine in barth solution at different pHs (6.3 and 7.3). Spermine potentiation was assessed in the presence 5 mM glutamate and 100 µM glycine. Under these conditions, GluN2Bwt-subunits were completely saturated, though both mutants were only at their ~$EC_{50}$. We therefore note that the difference in $I_{basal}$ between the three groups is an underestimation of the maximal $I_{basal}$, are therefore behind the lack of statistical significance. Arcaine sulfate was purchased from Alomone labs (Cat. # 14923-17-2). Ten mM solutions were made by diluting lyophilized Arcaine sulfate in barth solution at pH = 6.5. For competition assay, receptors were activated by 5 mM glutamate and 100 µM glycine solution, then blocked by 200 µM arcaine solution containing both agonists, followed by application of incrementing concentrations of spermine.

## Assessing potentiation by D-serine

GluNRs variants were expressed in *Xenopus* oocytes and were perfused with barth solution supplemented with 10 µM or 5 mM glutamate and 1-, 10-or 100 µM glycine. D-serine (sigma, Cat. #: 312-84-5) potentiation was examined by perfusing oocytes with barth solution supplemented with both co-agonists onto which we added constant D-serine concentration (100 µM). Amplitude fold change was calculated by normalizing the maximal current before and after exposure to D-serine.

## pH sensitivity

pH titration experiment was done by perfusing oocytes with barth solutions of different pHs, ranging from 6 to 8.3 (pH adjusted by NaOH), supplemented with 5 mM glutamate and 100 µM glycine.

## β-lactamase assay

HEK293T cells seeded in a 35 mm plates and transiently transfected with cDNA encoding hGluN1 and/or b-lac-GluN2B-wt/G689C/G689S/Kv4.2 using ViaFect (Promega). We used large dishes to obtain sufficient amount of cells for assessing 14 different conditions (amount of DNA used to transfect each group: 1 µg-hGluN2B-*wt*-β-lac, 1 µg hGlun1a + 1 µg hGluN2B-*wt*-β-lac, 1 µg hGlun1a + 1 µg hGlun2B-*wt*-β-lac/G689C-β-lac /G689S- β-lac, 1 µg hGlun1a + 1 µg hGluN2B-*wt*-β-lac + 0.1 µg hGluN2B-G689C/S/Kv 4.2, 1 µg hGlun1a + 1 µg hGluN2B-*wt*-β-lac + 0.5 µg hGluN2B-G689C/S/Kv 4.2, 1 µg hGlun1a + 1 µg hGluN2B-*wt*-β-lac + 1 µg hGluN2B-G689C/S/Kv 4.2). Six hours after transfection $20*10^4$ cells were seeded on PDL covered 96-well plates to which we added two different drugs (APV- 2 µM; MK-801–200 µM). Cells transfected with β-lac-GluN2B-*wt* alone were used to determine background absorbance and as a negative control for surface β-lactamase activity. Additionally, cells transfected with Kv4.2 (in different quantities) along with constant amounts of hGlun1a + hGluN2B-*wt*-β-lac were used as control for competition over translation machinery. Four-eight wells were seeded for each condition in each experiment. Twenty-four hrs following transfection, cells were washed with 200 µL Hank's Balanced Salt Solution (HBSS) supplemented with 10 mM HEPES. For measurements, we added 100 µL of 100 mM nitrocefin (Millipore - CAS 41906-86-9) in HBSS solution with 10 mM HEPES. The absorbance at 486 nm was measured every minute for 30–60 min at 30°C degrees by a plate reader. β-lactamase activity was determined from the slope of linear fit of the data.

## Structure modeling

Structural models of GluN2B LBD bearing G689C or G689S mutations generated using Schrodinger's Maestro 11.2. The glutamate-bound LBD of rat GluN2B (residues 403–539 and 651–802, PDB 4PE5 [*Karakas and Furukawa, 2014*]) was used as a template. All the structures were prepared using the Protein Preparation Wizard (Schrödinger Release 2021–1: Protein Preparation Wizard; Epik, Schrödinger, LLC, New York, NY, 2021) as implemented in Schrodinger's Maestro 11.2. This protocol adds missing hydrogen atoms considering a pH value of 7.2 ± 1.0 and optimizes the hydrogen bond network. Next, energy minimization was performed using MacroModel (Schrödinger Release 2021–2: MacroModel, Schrödinger, LLC, New York, NY, 2021) with the OPLS3e forcefield

and Polack-Ribiere Conjugate Gradient (PRCG) algorithm. Minimization was stopped either after 2500 steps of minimization or after reaching a convergence threshold of 0.05 kcal/mol. Graphic representation was done by PyMOL software.

## Data analysis

Data were analyzed by Clampfit software (Molecular Devices, USA) and were fitted using Sygmaplot 11 (Systat software, inc) to Hill's equation (*Equation 1*) from which we extracted $EC_{50}$ values. In co-expression experiments, results were best fitted by Michaelis-menten-like equations (*Equation 2* and *Equation 3*), assuming two affinity states, as previously described (*Ben-Chaim et al., 2003*). $IC_{50}$ values for $Mg^{+2}$ and proton inhibition were extrapolated by fitting the results to Hill's equation (*Equation 4*). $\tau_{off}$ values for apparent open probability were extrapolated by fitting the result to a mono-exponential function (*Equation 5*). $t_{10-90\%}$ results were obtained by placing cursors at beginning and end of responses, then automatically assessed by the Clampfit software (under statistics, positive going). mEPSCs were detected off-line using pCLAMP 10's template search. Briefly, 5–10 'minis' were identified and selected by user, followed by an automated search. All automatically identified minis were validated by user. Decay kinetics ($\tau_{off}$) were assessed by fitting responses by a mono-exponential function (*Equation 5*).

$$\text{Response} = 1/(1 + [(\text{glutamate})/\text{EC50}]^{nH}) \tag{1}$$

$$\gamma = \frac{B_{max} * X}{K_d^H + X} + \frac{B_{max} * X}{K_d^L + X} \tag{2}$$

$$\gamma = \frac{B_{max} * X^2}{\left(K_d^H + X\right)^2} + \frac{B_{max} * X^2}{\left(K_d^L + X\right)^2} \tag{3}$$

$$\text{Response} = \text{minimum} + ((1 - -\text{minimum})/1 + [(\text{Mg}^{+2}\text{orpH}/\text{IC}_{50}]^{nH})) \tag{4}$$

$$f(t) = \sum_{i=n}^{n} A_i e^{\frac{-t}{\tau_i}} + c \tag{5}$$

## Statistical analysis

All data are presented as mean ± SEM. Number of cells are indicated by n, whereas number of experiments by N. Statistical significance (Sigmaplot 11) was obtained by one-way ANOVA for multiple group comparison with post hoc Tukey test. *, $p < 0.05$; **, $p < 0.01$ and ***, $p < 0.001$; n.s., non-significant. Kruskal-Wallis ANOVA on ranks was used for assessing neuronal data (specifically, Rm, mini ratios and amplitudes; *Figure 8* and *Figure 8—figure supplement 2*). Significance of cumulative distributions was determined by Kolmogorov-Smirnov- (Sigmaplot 11).

## Acknowledgements

We thank the GRIN disorder research foundation (GDRF) for providing clinical data on the G689C patient and for the *GRIN* families. The research submitted is in partial fulfillment for a doctoral degree for SK and magister's degree for AA. **Funding—** Support was provided by the Israel Science Foundation (SB; 1096/17) and by TEVA pharmaceuticals scholarship (SK; PR783187).

## Additional information

### Funding

| Funder | Grant reference number | Author |
| --- | --- | --- |
| Israel Science Foundation | 1096/17 | Shai Berlin |
| Teva Pharmaceutical Industries | PR783187 | Shai Kellner |

The funders had no role in study design, data collection and interpretation, or the decision to submit the work for publication.

## Author contributions

Shai Kellner, Data curation, Software, Formal analysis, Visualization, Writing - original draft, Writing - review and editing; Abeer Abbasi, Data curation, Formal analysis, Investigation, Writing - review and editing; Ido Carmi, Data curation, Formal analysis, Investigation; Ronit Heinrich, Data curation, Project administration, Writing - review and editing; Tali Garin-Shkolnik, Tova Hershkovitz, Katrine M Johannesen, Rikke Steensbjerre Møller, Resources; Moshe Giladi, Data curation, Formal analysis; Yoni Haitin, Formal analysis; Shai Berlin, Conceptualization, Resources, Data curation, Software, Formal analysis, Supervision, Funding acquisition, Validation, Investigation, Visualization, Methodology, Writing - original draft, Project administration, Writing - review and editing

## Author ORCIDs

Shai Berlin (iD) https://orcid.org/0000-0002-5153-4876

## Ethics

Animal experimentation: All Experiments were approved by the Technion Institutional Animal Care and Use Committee (permit SB, no. IL-129-09-17).

## Decision letter and Author response

Decision letter https://doi.org/10.7554/eLife.67555.sa1
Author response https://doi.org/10.7554/eLife.67555.sa2

# Additional files

## Supplementary files

• Source code 1. Systematic replacement of all residues of the LBD by all other amino acids.

• Supplementary file 1. Validated *GRIN2B* variants. Description of output, functional validation and clinical pictures for reported *GRIN2B* variants. ATD- amino terminal domain; LBD- ligand binding domain; CTD- carboxy terminal domain; GoF and LoF- gain and loss-of-function, respectively.

• Supplementary file 2. Published and unpublished *GRIN2B* variants. >500 published and unpublished variants found in *GRIN2B* patients. Search was performed in PubMed, ClinVar and CFERV databases.

• Supplementary file 3. ΔΔG estimation for 5282 substitutions in LBD reveals that glycine and serine are vulnerable residues in GluN2B. a. ΔΔG values for 5282 substitutions in the S1/S2 segments (positions 403–539, 651–802) of GluN2B. Stabilizing mutations are represented by ΔΔG>0 (blue box), whereas destabilizing (ΔΔG< −0.5) are marked by red box. b. Most mutations within LBD are destabilizing (~60%; red), with the smallest fraction represented by stabilizing mutations (~16%; blue). Mutations with minor or no effect are noted by white color. c. List of 18 validated LBD mutations in GluN2B and their estimated ΔΔG. Point mutations resulting in stop codons are noted as stop. d. Overlay between mutations shown in c. (dashed arrows) and mean ΔΔG; calculated for each residue by averaging all its possible substitutions for that residue. Red- destabilizing, blue-stabilizing; gray-neutral effect. e. Surface view of the LBD shows that stabilizing residues (blue, taken from d) are located on the outer surface of the LBD. f. Potential effect of all amino acids if these were substituted anywhere in the LBD. This consists of averaging the effect of each substitution across the entire LBD. Analysis highlights glycine (G) and serine (S) as the most destabilizing mutations, whereas methionine (M) as the most stabilizing.

• Transparent reporting form

## Data availability

All data generated or analysed during this study are included in the manuscript and supporting files. Raw data is deposited at Dryad (https://doi.org/10.5061/dryad.1jwstqjv3).

The following dataset was generated:

| Author(s) | Year | Dataset title | Dataset URL | Database and Identifier |
|---|---|---|---|---|
| Kellner S, Abbasi A, Heinrich R, Garin-Shkolnik T, Hershkovitz T, Johannesen KM, Møller RS, Berlin S | 2021 | Two de novo GluN2B mutations affect multiple NMDAR-functions and instigate severe pediatric encephalopathy | http://dx.doi.org/10.5061/dryad.1jwstqjv3 | Dryad Digital Repository, 10.5061/dryad.1jwstqjv3 |

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
