## [Decision Letter]

**Acceptance summary:**

This interesting study characterizes NMDA receptor mutants in regard to ion channel physiology and to functional consequences from clinical impairments of two pediatric cases. A strength of the study is that missense mutations are analyzed with respect to ion channel receptor function. The mechanism of a dominant negative effect on NMDA receptor function is proposed. A combination of modeling and electrophysiological methods is used to support the key conclusions of the paper.

**Decision letter after peer review:**

Thank you for submitting your article "Two de novo GluN2B mutations affect multiple NMDAR-functions and instigate severe pediatric encephalopathy" for consideration by *eLife*. Your article has been reviewed by 3 peer reviewers, and the evaluation has been overseen by a Reviewing Editor and Huda Zoghbi as the Senior Editor. The reviewers have opted to remain anonymous.

The reviewers have discussed their reviews with one another, and the Reviewing Editor has drafted this letter to help you prepare a revised submission. Detailed comments are below.

Essential revisions:

The study was considered to be well done and to be of interest. Positive comments were also raised about the nature and details of the findings. Several major reservations were raised. There have been many characterizations of NMDAR variants that have been published. Given the two missense mutations in the agonist binding site of GluN2B, the conclusions should be strengthened by additional pharmacological or translational insights to delineate the novelty and impact of the results.

1) Either a novel insight into the function of NMDARs or a new clinical insight would greatly enhance the impact of the manuscript. There are many papers being generated on this topic that draw correlations between functional characterizations and clinical phenotypes. We are not certain that they provide new insights into the etiology of the disease phenotype or potential treatment options.

2) One point to address is the nature of the dominant negative mechanism. A discussion of significance should make clear the novelty of the dominant negative mechanism of suppression of receptor currents by the mutations studied here. How is this mechanism distinguished from other disability-causing mutations that have been reported elsewhere. The NMDA receptor does not open unless all four LBDs are occupied. Therefore, the incorporation of one mutant NR2B into the hetero-tetramer can block function. So mutant NR2B Is dominant over wild type NR2B, in the same tetramer. Is this novel or are most NMDA receptor mutations like this?

*Reviewer #1:*

NMDA receptors are glutamate-gated ion channels that make significant contributions to a variety of brain functions. Recently, de novo missense mutations have been identified in genes encoding NMDA receptor subunits and are often associated with severe clinical phenotypes. How these missense mutations lead to clinical phenotypes is unknown and one way to address this is to characterize their effect on receptor function. In the present manuscript, the authors characterize the effect of two missense mutations at a glycine in the agonist binding pocket of the GluN2B subunit. The authors find that the mutations have different effects on receptor function and propose that these differences may contribute to variations in their associated clinical phenotypes.

The manuscript is a positive characterization of two missense mutations at a site in the agonist binding pocket of the GluN2B subunit. The data are quite reasonable and are displayed appropriately. The general conclusions are appropriate.

Potential impact of the manuscript. While a nice characterization, many of these have been done and it is not clear that the results in the present manuscript provide a high impact insight. Perhaps it is there but the authors need to do a better job of demonstrating that they have made a significant new discovery about NMDA receptors, either in terms of structure-function or clinical aspects. Many characterizations of missense mutations in NMDA receptors have been done and have provided new insights into how receptors carry out their biological function (e.g., Odgen et al., 2017, PLoS Genetics, Amin et al., 2018, Nat Comm; Fedele et al., 2018, Nat Comms) or how they might function at synapse (e.g., Liu et al., 2017 J. Neurosci.). Other pathways are available to provide impact, e.g., providing some new therapeutic insight or potentially how they affect signaling in vivo but these are absent in the present version.

Again, perhaps there is buried in the manuscript some new insight but at present it feels like a number of observations (e.g., that has a very strong effect on glutamate potency or have effects on pH sensing) with no clear advance in terms of mechanism. Also in terms of relating the observed effects of the missense mutations on receptor function to clinical phenotypes is always challenging and is really just speculation unless a large number are done.

*Reviewer #2:*

This is a well-written manuscript in which the authors have identified two patients with missense mutations in the GRIN2B gene, substituting a highly conserved glycine residue. They find the mutations produce marked reductions in glutamate potency, one showing the largest reduction in EC50 reported for GRIN2B-mutations (~2000-fold reduction). Variants readily multimerize with the GluN2Bwt- subunits, promote their surface expression and exert a dominant- negative effect on receptor function. Interestingly, one variant entirely fails to respond to spermine, a specific GluN2B-potentiator, at physiological pH. The manuscript provides important contribution to the understanding of functional consequences of two specific mutations of the GRIN2B gene. As such the manuscript is a great example of how to integrate knowledge on basic receptor function and personalized medicine. This is indeed achieved by combining sophisticated modeling and electrophysiological techniques that have permitted addressing of important aspects of receptor function which has been investigated in depth by a large variety of experiments that were well executed. Because of the clinical relevance of the study, the research and clinical community should learn from this manuscript and be interested in it. This reviewer could not identify any weakness in the manuscript.

*Reviewer #3:*

Kelner et al. have studied the effects of naturally occurring mutations in the ligand binding domain of the GluN2B subunit of the NMDA receptor. Two mutations at the same residue, G689S and G698C, were identified in two pediatric patients with severe intellectual disability and delayed neural development. The authors describe the clinical manifestations of these mutations and investigate the molecular and physiological consequences through expression in oocytes and mammalian cells.

The GluN2B crystal structure suggested that the mutations would occlude glutamate binding, and indeed, while the EC50 (glutamate concentration for half maximal current) of the wt was in the μM range, the EC50 was 1000 x higher for 2B-G689S and 2000 x higher for 2B-G698C, with both in the millimolar range. This indicates reduced affinity of glutamate for the mutant receptors leading to loss of function, although at saturating glutamate concentrations (10mM), Imax for the mutants and wt were similar.

Experiments in 293 cells showed that surface expression of GluN2B-G689S was similar to wt, and GluN2B2B-G689C was 45% of this value. Both variants increased the surface expression of wt subunits, indicating that both mutants could multimerize with wt and GluN1a to form surface-expressed heterotetramers. Expression of increasing levels of mutant with an excess of wt progressively depressed currents and increased KD values, supporting co-assembly of mutant and wt into heterotetramers resulting in loss of function through a strong dominant negative effect on glutamate affinity, but a much lower effect on co-assembly and surface expression.

Analysis of dose-response curves indicated that co-expression of wt and mutant led to the formation of two receptor populations, one wt and the other incorporating a mutant receptor.

In an attempt to find a pharmacologic treatment that overcame the diminished currents of the mutants, the authors treated oocytes expressing hetero-tetramers with spermine, a GluN2B-specific potentiator. However under conditions for which spermine potentiates wt receptors, spermine poorly potentiated the mutants. Furthermore, decreasing the pH, which greatly increased spermine potentiation of wt, only modestly increased mutant receptor currents in oocytes. Experiments with a competitive antagonist of spermine indicated that the spermine binding site was intact in the mutants. Further, the mutants were more resistant to proton inhibition, which suggested that residue 689 is part of the proton sensing domain.

Other experiments showed that while D-serine, another possible potentiator, increased currents of wt and of the mutants, this increase was eliminated in the presence of the co-agonist, glycine, indicating D-serine simply augments filling the glycine site in GluN1a, rather than acting as a true potentiator.

Finally, the current reduction by GluN2B-G689C seen in oocytes was reproduced in the mammalian 293 cell line, confirming the relevance of the oocyte studies.

The paper presents a comprehensive analysis of the changes in physiological and cell biological properties of two GluN2B subunits with LBD mutations that are associated with intellectual disability and delayed development, and in one case, epileptic seizures. The data support a mechanism of dominant negative inhibition of NMDA receptor function, in which the mutant subunit assembles with wt GluN1a and wt GluN2B to form tri-heterotetrametric receptors that traffic to the plasma membrane but fail to conduct because they have very low affinity for glutamate.

In addition to the graphic presentation of the data in Figures, the data is also presented numerically in Tables in a comprehensive form. Also summarized in Tables are the clinical assessments of the two proband patients from whom the mutants were isolated.

The patient with the G689S mutation also had mutations in SLC6A8 gene (sodium- and chloride-dependent creatine transporter 1) and CACNA1A gene (Calcium Voltage-Gated Channel Subunit Alpha1A). It is not known whether these mutations contributed to the clinical phenotype. Similar data are not available for the G689C mutation patient.

The paper provides considerable insight into the mechanism by which GluN2B mutations exert a strong, dominant negative effect on NMDA receptor function. The paper does not address how these mutations affect function in neurons, either in vivo or in vitro.

This article presents a set of well-chosen and well executed experiments that supports a dominant-negative mode of action of mutant NMDA receptor GluN2B subunits that cause severe intellectual disability.

Although not necessary for this submission or for drawing the main conclusions of the article, experiments in neurons, either in vivo in genetically modified mice, or in cultured neurons, would enhance this report. Experiments in oocytes and heterologous cells, although providing valuable information about receptor assembly, trafficking and electrophysiological function, cannot take into account factors that arise only in vivo in a neuronal context. These include the developmental patterns of expression of the various NMDA receptor subunits and binding proteins, and neuron-specific interactions including ones at synapses. Also, the effects of the mutations on synaptic plasticity, such as LTP/LTD, cannot be assessed.

The conclusions of a number of the experiments depend on expressing wt and mutant receptors at comparable or other predetermined levels, and this can be affected by the quality of oocyte-injected mRNA, differences in expression efficiency and protein stability etc. Indeed, the mutant receptors are not expressed at the surface with the same efficiency and have differences in expression and/or stability. The authors should discuss more fully how these parameters were addressed.

The authors discuss the effects of the mutations on the stereochemistry of the ligand binding site occupied by glutamate. But because the mutations occlude binding glutamate, it would be interesting to consider the LBD structure of the mutants in the unliganded state, if the appropriate crystal structure were available. This might suggest a pharmacologic (small molecule) means for closing the clamshell and opening the pore of the mutant, even in the absence of bound glutamate.

The speculation about the effects of water access on deactivation kinetics (p 5), which turned out to be incorrect, interrupts the flow of the Results section and might be considered better in the Discussion.

The manuscript has a significant number of typographical errors and a large number of grammatical errors or non-colloquial phrasings. It should be revised carefully by a native English speaker.

---

## [Author Response]

Essential revisions:The study was considered to be well done and to be of interest. Positive comments were also raised about the nature and details of the findings. Several major reservations were raised. There have been many characterizations of NMDAR variants that have been published. Given the two missense mutations in the agonist binding site of GluN2B, the conclusions should be strengthened by additional pharmacological or translational insights to delineate the novelty and impact of the results.1) Either a novel insight into the function of NMDARs or a new clinical insight would greatly enhance the impact of the manuscript. There are many papers being generated on this topic that draw correlations between functional characterizations and clinical phenotypes. We are not certain that they provide new insights into the etiology of the disease phenotype or potential treatment options.

We thank the reviewers for highlighting the issue of novelty. We now realize to have clearly downplayed the importance of our findings. We would therefore like emphasize that our study not only provides description and characterization of two novel GRIN2B variants, it also provides several new insights into NMDAR function, dominant effect of variants over tri-heteromeric channels and effect in neurons. These are now specifically highlighted throughout the text.

Briefly, the novelties we report are:

1. We describe two mutations that lead to the most extreme cases of reduction in glutamate potency of the GluN2B subunit (and only second to a *GRIN2A* mutation). We therefore expected these mutations to mirror effects of mutations that completely abolish expression of the subunit (e.g., truncation mutations^1^). This would suggest that the disease could be associated with cases of haploinsufficiency (as commonly suggested in literature^2–4^). Instead, we observe the opposite. First G689S expresses as well as the native subunit and, although G689C expresses less, both variants significantly reduce glutamate potency of mixed channels (Figures 2 and 3). Additionally, both variants promote the expression of 2Bwt subunits (Figure 3). In neurons, the variants affect synaptic currents (significantly alter frequency and amplitude of miniature EPSCs; Figure 8 and Figure 8—figure supplement 2). These are best explained by a very strong dominant-negative effect (*and* see next reply specifically addressing the novelty of the dominant negative effect).

2. We provide new insights pertaining to glutamate-affinity and expression levels. Briefly, high affinity is not essential for proper membrane trafficking. This is very different than most LoF mutations in the LBD and argues against classification of the latter as haploinsufficiency (now explicitly noted in page 11). In fact, we deem this as a major issue as the medical community often confuses between these terms (For an explicit example see our reply for query #2).

3. We discover that the G689 residue is involved in proton sensing of the GluN2B-subunit.

4. We present new simulations showing that residues that protrude towards the glutamate and shift its position are likely behind the significant decrease in glutamate-affinity (Figure 1 and Figure 1—figure supplement 1).

5. Moreover, we present ∆∆G predictions for ALL possible mutations in the LBD (consisting of 5282 mutations; we provide the scripts and graphical analysis in Suppl. File 3). These suggest that most substitutions (~60%) within the LBD destabilize its structure; providing plausible explanation why most LBD mutation lead to reduced affinity to glutamate (i.e, LoF). The data also show that substitutions to glycines or serines are the most destabilizing. Interestingly, the same may apply inversely, namely removal of glycine and its substitution by other (larger) residues should be highly disfavorable for receptor function. In support, glycine residues are suggested to serve as essential hinges in GRIN2B and their mutagenesis causes severe channel dysfunction^5^, including the two cases shown here (pages 18-19)

Per the second part of the question (potential treatment), we would first like reiterate that this was our primary goal and the reason behind the scrutiny of specific and non-specific channel potentiators (spermine and D-serine, respectively). These failed to enhance receptor function (but provided valuable insights into proton sensitivity). Notably, whereas most therapeutics target GoF mutations, and therefore employ channel blockers (most commonly FDA-approved pan-NMDAR blocker memantine, e.g., ^3,6–10^, pages 3-4)., there are very few openers (e.g., ^11–13^) and even fewer GluN2B-selective.

Nevertheless, to try and develop a potential treatment, we wanted to explore gene editing techniques, specifically base editing^14^ (see details below). We examined all possible substitutions that can be made:

WT – p.689 – GGC,

G689C – p.689 – TGC

G689S – p.689 – (AGC)

C•G to T•A base editors (CBEs)-

G689C – p.689 – TGT = maintains Cysteine mutation

G689S – p.689 – AGT = maintains Serine mutation

A•T to G•C base editors (ABEs)

G689C – p.689 – TGC – none-modifiable

G689S – p.689 – GGC – Glycine **restores to normal**

C•G to G•C base editors^15^

G689C – p.689 – TGG – modifies to Tryptophan (W)

G689S – p.689 – AGG – modifies to Arginine (R)

Whereas only G689S can be restored to normal (i.e., glycine) by using A•T to G•C base editors (ABEs), G689C cannot. G689C can only be modified to tryptophan using C•G to G•C base editors^15^ (which would produce an arginine if applied onto G689S). We produced both of these variants and tested in oocytes and find that both yield very low expression levels/complete loss of function. We would be happy to include these results, even if negative should the reviewers deem it interesting.

Lastly, we address the comment about genotype-phenotype correlations by several ways. First, we provide an extensive analysis of the literature, collating *all r*eported GRIN2B mutations (using PubMed, ClinVar and CFERV databases; described in text- page 17 and shown in Suppl. File 2), as well as focus on LBD mutations (Suppl. Table 1). Unfortunately, there are too few LBD mutations with reported characterizations (<20), and not all mutations show a clear binary effect (e.g., I655F mutation acts both as LoF mutation – instigating ~3-fold reduction in glutamate EC_50_, as well as a GoF mutation –by reducing Mg^2+^-sensitivity) from which we can draw any firm conclusions (Suppl. File 3). We also examined several correlations with the various clinical phenotypes though fail to find any clear associations. These observations are not completely unexpected (and the reason why we have initially left them out of the manuscript), as similar analyses have been previously performed and they all provide a very limited genotype-phenotype associations (e.g., ^2,7,16^; see below for specific details). Resultantly, there is lack of formal diagnostic criteria based on clinical phenotype for GRIN2B-related disorders (^17^). We now note all these in discussion (pages 16-17), in particular the current consensus regarding the strong evidence that mutations in GRIN2B are more associated with developmental delays and autism spectrum disorder, whereas GRIN2A more associated with epilepsy (^2^ and “Clinical and genetic spectrum of GRIN2A and GRIN2B variants.” Dr. Johannes Lemke. University of Leipzig. Presented at 2019 CFERV Conference on GRIN Variants. Atlanta. 2019).

Details:

A recent analysis of the largest GRIN2B patient cohort (86 patients) found no clear association between: (1) effect of mutation (GoF or LoF), (2) extent of effect (how much gain or loss compared to wt), (3) localization of the mutations and clinical phenotype (^6^ and Clinical and genetic spectrum of GRIN2A and GRIN2B variants.” Dr. Johannes Lemke. University of Leipzig. Presented at 2019 CFERV Conference on GRIN Variants. Atlanta. 2019). The only significant correlation obtained was between variant class (i.e., missense or truncation) and intellectual outcome (mild to moderate moderate vs severe ID) (Fisher’s exact test, p=0.0079) with truncation carriers tending to present mild/moderate intellectual disability. This is not completely unexpected as this appears to be the only recurring theme for GRIN2B mutations (as we have mentioned in the text) owing to its embryonic expression. With regard to GRIN2A, analysis of a larger cohort (248 patients) found only two distinct phenotype groups that could be explained by the location and effect (gain or loss – but not size of effect) of the mutations^7^. More precisely, the authors show that most mutations in the TMDs (and there are 4 TMDs) or linkers are associated with broad developmental and epileptic encephalopathy phenotypes and these mutations are also typically associated with GoF of the receptors (though to very different extents), whereas mutations in the two large extracellular domains (amino terminal domain and ligand binding domain) are associated with speech abnormalities and/or seizures with mild to no ID only; and these typically instigate LoF^7^ (these are now mentioned in page 17).

2) One point to address is the nature of the dominant negative mechanism. A discussion of significance should make clear the novelty of the dominant negative mechanism of suppression of receptor currents by the mutations studied here. How is this mechanism distinguished from other disability-causing mutations that have been reported elsewhere. The NMDA receptor does not open unless all four LBDs are occupied. Therefore, the incorporation of one mutant NR2B into the hetero-tetramer can block function. So mutant NR2B Is dominant over wild type NR2B, in the same tetramer. Is this novel or are most NMDA receptor mutations like this?

We agree with the reviewers that the dominant-negative effect should be further discussed, as it is unique as it diverges from the effect of other reported mutations. We now provide these details in pages 10-11 and in the discussion. We highlight the fact that a dominant-negative effect in *GRINs* is not commonly reported. In support, only a handful of reports (three) explicitly describe or note a dominant-negative effect for three *GRIN* variants (GRIN1^18^, GRIN2A^19^ and GRIN2B^20^). In two reports (^18,19^) the authors interpret reduced current amplitudes (~50%) as an indication for dominant negative effect, whereas the third shows that a single copy of GluN2B‐N616K produces a dominant reduction in Mg^2+^-block similar to channels including two copies of the variant^20^. However, most reports examining other *GRIN* mutations do not describe dominance, even though we expect the least functional subunit to be the limiting factor in mixed channels. For instance, a recent report examining eight different *GRIN* variants (M2-pore mutations) shows that mixed-channels exhibit very mild reduction in Mg^2+^ IC_50_, with values corresponding to values of the wt channels^20^. Another report examining mixed-channels containing GluN2Awt and 2A-P552R^21^ shows that, whereas 2A-P552R significantly alters stability of the pore when it is found in two copies per channel, it fails to do so when mixed with GluN2Awt. Very similar observations are reported for mixed channels bearing 2Bwt and the GluN2B-E413G variant in which the EC_50_ is not dominated by the low affinity subunit ^16^. Thus, while a dominant negative effect is somewhat intuitive – as all LBDs of NMDARs need to be liganded for full channel opening^22–24^ – it is not commonly observed in *GRINs*, especially not LBD mutations of *GRIN2B*.

Notably, following an extensive re-review of the literature, we found two additional reviews (specifically^25,26^) that cite the reports mentioned above, and notice that they also cite additional ones. The reviews interpret the findings of the added reports as proof for dominant negative. However, close scrutiny of the added citations shows that the original reports do not show a dominant negative effect, nor suggest it in their original manuscripts Chen et al. ^26^ refers readers to references #5,6-11, 13,16 and 18; Poot M^25^, refers to Lemke 2016 and Platzer 2017. Thus, we still maintain that we show a unique dominant negative effect.

Significance of dominant negative findings – Our findings diverge from reports that examined the effect of variants on mixed channels (i.e., channel made of one copy of a *wt*-subunit mixed with another copy of the variant). First, previous reports reach this conclusion by relying on a single observation whereby mixed channels exhibit reduced (~50%) current amplitudes. However, as reduced currents can result from multiple reasons, such as expression. This motivated us to address this by multiple means. Indeed, we do not rely on a single DNA-mixture and current amplitude, rather explore different channel stoichiometries (by employing a highly established method of mRNA titrations, e.g., ^27–29^) and assess current amplitudes, expression levels and, importantly, glutamate potency. These ultimately lead to the likely interpretation that mixed channels that exhibit normal current amplitudes (supported by expression assays, Figure 3) contain the *wt* subunits, but the very reduced EC_50_ results from the limiting effect of the variants. We go on to confirm that our results do not reflect two distinct populations of channels (Figure 4). Notably, this is different than previous reports (see ^16^). Thus, we do not simply suggest a dominant negative, we explore it by numerous means and provide strong evidence to support it. Finally, though severe LoF are commonly equated to cases of haploinsufficiency (e.g., by null mutations and early stop codons), we show that this is not the case for these two variants. We now provide these details in the discussion.

Reviewer #1:NMDA receptors are glutamate-gated ion channels that make significant contributions to a variety of brain functions. Recently, de novo missense mutations have been identified in genes encoding NMDA receptor subunits and are often associated with severe clinical phenotypes. How these missense mutations lead to clinical phenotypes is unknown and one way to address this is to characterize their effect on receptor function. In the present manuscript, the authors characterize the effect of two missense mutations at a glycine in the agonist binding pocket of the GluN2B subunit. The authors find that the mutations have different effects on receptor function and propose that these differences may contribute to variations in their associated clinical phenotypes.The manuscript is a positive characterization of two missense mutations at a site in the agonist binding pocket of the GluN2B subunit. The data are quite reasonable and are displayed appropriately. The general conclusions are appropriate.Potential impact of the manuscript. While a nice characterization, many of these have been done and it is not clear that the results in the present manuscript provide a high impact insight. Perhaps it is there but the authors need to do a better job of demonstrating that they have made a significant new discovery about NMDA receptors, either in terms of structure-function or clinical aspects. Many characterizations of missense mutations in NMDA receptors have been done and have provided new insights into how receptors carry out their biological function (e.g., Odgen et al., 2017, PLoS Genetics, Amin et al., 2018, Nat Comm; Fedele et al., 2018, Nat Comms) or how they might function at synapse (e.g., Liu et al., 2017 J. Neurosci.). Other pathways are available to provide impact, e.g., providing some new therapeutic insight or potentially how they affect signaling in vivo but these are absent in the present version.Again, perhaps there is buried in the manuscript some new insight but at present it feels like a number of observations (e.g., that has a very strong effect on glutamate potency or have effects on pH sensing) with no clear advance in terms of mechanism. Also in terms of relating the observed effects of the missense mutations on receptor function to clinical phenotypes is always challenging and is really just speculation unless a large number are done.

We thank the review for the positive comments on our manuscript and understand his/her comment about list of observations. We would like to emphasize that we have not simply tested random features, rather the results of one observation led us to test the next. More specifically, our hypothesis regarding lower glutamate potency (based on location of mutation, simulations and literature^3^) led to the assessment of glutamate affinity (but not other features related to other domains, such as amino terminal domain). The smaller currents motivated evaluation of expression. Lastly, our aspirations to find a potential treatment led us to explore channel potentiators, which in turn led us to uncover the variants’ diminished proton-sensitivity and to elaborate on the *agonistic* actions of D-serine, rather than *potentiator*.

Reviewer #3:[…] Although not necessary for this submission or for drawing the main conclusions of the article, experiments in neurons, either in vivo in genetically modified mice, or in cultured neurons, would enhance this report. Experiments in oocytes and heterologous cells, although providing valuable information about receptor assembly, trafficking and electrophysiological function, cannot take into account factors that arise only in vivo in a neuronal context. These include the developmental patterns of expression of the various NMDA receptor subunits and binding proteins, and neuron-specific interactions including ones at synapses. Also, the effects of the mutations on synaptic plasticity, such as LTP/LTD, cannot be assessed.

We now provide results from cultured rat hippocampal neurons in which we have overexpressed 2B, G689C or G689S (Figure 8 and Figure 8—figure supplement 2, page 15). Briefly, overexpression of the variants causes a strong reduction in synaptic GluNR-events: neurons overexpressing the variants showed a strong reduction in the frequency of mini_NMDAR_, without any effect on the frequencies of mini_AMPAR._ Moreover, overexpression of G689S (but not G689C) reduces amplitudes of mini_NMDAR_, with a concomitant increase in the amplitude of mini_AMPAR_. Lastly, mini_NMDARs_ from neurons overexpressing the variants display faster deactivation kinetics than control. Together, these results demonstrate that expression of the variants in hippocampal neurons prompts a pronounced effect on synaptic GluNRs. The reduction in the frequency of mini_NMDAR_ – in combination with the unaffected frequency of mini_AMPAR_ – rules-out loss of excitatory synapses (as may be instigated by other variants, e.g., GluN2B-S1413L^31^). Moreover, G689S dual effect on amplitudes of both mini-types suggests that the ‘silencing’ of GluNRs (by the dominant negative effect) induces compensatory mechanism that drive increases in synaptic GluARs^32^. Lastly, acceleration of τ_off_ due to overexpression of the variants reveals that the remaining minis are more associated with the faster deactivating GluN2A subunits^33^. Notably, all of these effects are highly consistent with our dominant negative observations (see Figure 4) and with results obtained from animal model bearing a GluN2B LoF mutation^34^.

The conclusions of a number of the experiments depend on expressing wt and mutant receptors at comparable or other predetermined levels, and this can be affected by the quality of oocyte-injected mRNA, differences in expression efficiency and protein stability etc. Indeed, the mutant receptors are not expressed at the surface with the same efficiency and have differences in expression and/or stability. The authors should discuss more fully how these parameters were addressed.

We now elaborate on these in methods. Briefly, mRNA quality is systematically assessed by electrophoresis and diluted mRNA is aliquoted, and stored at -80 degrees (as previously reported^28,35^). Thus, the same batch of mRNA serves in multiple experiments. Moreover, we also note that we do not record from dying/unhealthy oocytes (assessed by appearance, resting potentials and leak currents). All experiments are repeated multiple times. In the particular case of the mixed groups experiments (Figure 4), the experiments themselves serve as control for mRNA quality. All groups are injected with the same GluN1a mRNA. Moreover, every experiment consists of multiple groups, including control groups that express the variants or *wt* subunits on their own, not to mention the groups in which one mRNA is excessively expressed over the other. (e.g., G689C mRNA >> wt mRNA; 16:1). In cases these failed to provide any current, we did not analyze the experiment. However, as the reviewer rightly notes, when the variants are mixed with the 2Bwt-subunit, expression levels may vary – but only in the case of the G689C subunit (G689S expresses as well as 2Bwt). In mixed groups, we obtain large currents in all groups despite the differing mRNA stoichiometries. In the case of G689C, this is indicative of the expression of the wt subunits, but not in the case of G689S. The dramatic drop in EC_50_ (and complete lack of two populations assessed by various fitting modes), demonstrates that there are no purely-2B*wt* channels, rather most are mixed channels. Of note, this is one particular strength in our work over previous reports, namely employing mRNA titration for controlling protein levels is a (e.g., ^28,29,35^) as we could determine a bona-fide dominant negative effect that does not solely rely on current amplitudes (i.e., ^18,19^), which is prone to yield errors owing to the limitations mentioned by the reviewer.

The authors discuss the effects of the mutations on the stereochemistry of the ligand binding site occupied by glutamate. But because the mutations occlude binding glutamate, it would be interesting to consider the LBD structure of the mutants in the unliganded state, if the appropriate crystal structure were available. This might suggest a pharmacologic (small molecule) means for closing the clamshell and opening the pore of the mutant, even in the absence of bound glutamate.

We agree with the reviewer, but there are no crystal structures of this LBD (or other subunits) without a ligand. However, we now provide better simulations (along PDBs) of the LBD with the mutations (Figure 1).

The speculation about the effects of water access on deactivation kinetics (p 5), which turned out to be incorrect, interrupts the flow of the Results section and might be considered better in the Discussion.

We have re-written this part. We describe the general outcomes of the simulations, without going into detail about the holes, to lessen interruption.

The manuscript has a significant number of typographical errors and a large number of grammatical errors or non-colloquial phrasings. It should be revised carefully by a native English speaker.

The text has been substantially revised, edited and corrected by a native English speaker, as suggested. We hope this version reads better.

References:

1. Hu, C., Chen, W., Myers, S. J., Yuan, H. and Traynelis, S. F. Human GRIN2B variants in neurodevelopmental disorders. J. Pharmacol. Sci. 132, 115–121 (2016).

2. Myers, S. J. et al. Distinct roles of GRIN2A and GRIN2B variants in neurological conditions. F1000Research 8, (2019).

3. XiangWei, W., Jiang, Y. and Yuan, H. de novo mutations and rare variants occurring in NMDA receptors. Curr. Opin. Physiol. 2, 27–35 (2018).

4. Santos-Gómez, A. et al. Disease-associated GRIN protein truncating variants trigger NMDA receptor loss-of-function. Hum. Mol. Genet. 29, 3859–3871 (2020).

5. Amin, J. B., Leng, X., Gochman, A., Zhou, H.-X. and Wollmuth, L. P. A conserved glycine harboring disease-associated mutations permits NMDA receptor slow deactivation and high Ca 2+ permeability. Nat. Commun. 9, 3748 (2018).

6. Platzer, K. et al. GRIN2B encephalopathy: novel findings on phenotype, variant clustering, functional consequences and treatment aspects. J. Med. Genet. 54, 460–470 (2017).

7. Strehlow, V. et al. GRIN2A -related disorders: genotype and functional consequence predict phenotype. Brain 142, 80–92 (2019).

8. Gheța, I. et al. GRIN2A Variant in A 3-Year-Old – An Expanding Spectrum? Neurol. Int. 13, 184–189 (2021).

9. Amador, A. et al. Modelling and treating GRIN2A developmental and epileptic encephalopathy in mice. Brain (2020) doi:10.1093/brain/awaa147.

10. Fedele, L. et al. Disease-associated missense mutations in GluN2B subunit alter NMDA receptor ligand binding and ion channel properties. Nat. Commun. 9, 957 (2018).

11. Burnell, E. S. et al. Positive and Negative Allosteric Modulators of N-Methyl-D-Aspartate (NMDA) Receptors; Structure-Activity Relationships and Mechanisms of Action. J. Med. Chem. 62, 3–23 (2019).

12. Hackos, D. H. et al. Positive Allosteric Modulators of GluN2A-Containing NMDARs with Distinct Modes of Action and Impacts on Circuit Function. Neuron 89, 983–999 (2016).

13. Addis, L. et al. Epilepsy-associated GRIN2A mutations reduce NMDA receptor trafficking and agonist potency – molecular profiling and functional rescue. Sci. Rep. 7, 66 (2017).

14. Porto, E. M., Komor, A. C., Slaymaker, I. M. and Yeo, G. W. Base editing: advances and therapeutic opportunities. Nat. Rev. Drug Discov. 19, 839–859 (2020).

15. Kurt, I. C. et al. CRISPR C-to-G base editors for inducing targeted DNA transversions in human cells. Nat. Biotechnol. 39, 41–46 (2021).

16. Swanger, S. A. et al. Mechanistic Insight into NMDA Receptor Dysregulation by Rare Variants in the GluN2A and GluN2B Agonist Binding Domains. Am. J. Hum. Genet. 99, 1261–1280 (2016).

17. Platzer, K. and Lemke, J. R. GRIN2B-Related Neurodevelopmental Disorder. in GeneReviews (eds. Adam, M. P. et al.) (University of Washington, Seattle, 1993).

18. Lemke, J. R. et al. Delineating the GRIN1 phenotypic spectrum. Neurology 86, 2171–2178 (2016).

19. Endele, S. et al. Mutations in GRIN2A and GRIN2B encoding regulatory subunits of NMDA receptors cause variable neurodevelopmental phenotypes. Nat. Genet. 42, 1021–1026 (2010).

20. Li, J. et al. de novo GRIN variants in NMDA receptor M2 channel pore‐forming loop are associated with neurological diseases. Hum. Mutat. 40, 2393–2413 (2019).

21. Ogden, K. K. et al. Molecular Mechanism of Disease-Associated Mutations in the Pre-M1 Helix of NMDA Receptors and Potential Rescue Pharmacology. PLoS Genet. 13, (2017).

22. Berlin, S. et al. A family of photoswitchable NMDA receptors. *eLife* 5, e12040 (2016).

23. Wilding, T. J., Lopez, M. N. and Huettner, J. E. Radial symmetry in a chimeric glutamate receptor pore. Nat. Commun. 5, 3349 (2014).

24. Kussius, C. L. and Popescu, G. K. Kinetic basis of partial agonism at NMDA receptors. Nat. Neurosci. 12, 1114–1120 (2009).

25. Poot, M. Phenotypic Spectrum and Severity of Disease Depending on the Mutated Protein Domain of NMDA Receptor-Encoding Genes. Mol. Syndromol. 10, 127–129 (2019).

26. Chen, W. et al. GRIN1 mutation associated with intellectual disability alters NMDA receptor trafficking and function. J. Hum. Genet. 62, 589–597 (2017).

27. Peleg, S., Varon, D., Ivanina, T., Dessauer, C. W. and Dascal, N. Gαi Controls the Gating of the G Protein-Activated K^+^ Channel, GIRK. Neuron 33, 87–99 (2002).

28. Berlin, S. et al. A Collision Coupling Model Governs the Activation of Neuronal GIRK1/2 Channels by Muscarinic-2 Receptors. Front. Pharmacol. 11, (2020).

29. Katz, M. et al. Reconstitution of β-adrenergic regulation of CaV1.2: Rad-dependent and Rad-independent protein kinase A mechanisms. Proc. Natl. Acad. Sci. 118, (2021).

30. Pires, D. E. V., Ascher, D. B. and Blundell, T. L. mCSM: predicting the effects of mutations in proteins using graph-based signatures. Bioinformatics 30, 335–342 (2014).

31. Liu, S. et al. A Rare Variant Identified Within the GluN2B C-Terminus in a Patient with Autism Affects NMDA Receptor Surface Expression and Spine Density. J. Neurosci. 37, 4093–4102 (2017).

32. Sutton, M. A. et al. Miniature Neurotransmission Stabilizes Synaptic Function via Tonic Suppression of Local Dendritic Protein Synthesis. Cell 125, 785–799 (2006).

33. Paoletti, P., Bellone, C. and Zhou, Q. NMDA receptor subunit diversity: impact on receptor properties, synaptic plasticity and disease. Nat. Rev. Neurosci. 14, 383–400 (2013).

34. W, S. et al. Early correction of synaptic long-term depression improves abnormal anxiety-like behavior in adult GluN2B-C456Y-mutant mice. PLoS Biol. 18, (2020).

35. Berlin, S. et al. Gαi and Gβγ Jointly Regulate the Conformations of a Gβγ Effector, the Neuronal G Protein-activated K^+^ Channel (GIRK). J. Biol. Chem. 285, 6179–6185 (2010).